# Psychosocial Perceptions and Health Behaviors Related to Lifestyle During Pregnancy: A Cross-Sectional Study in a Local Community of Albania

**DOI:** 10.3390/healthcare14020172

**Published:** 2026-01-09

**Authors:** Saemira Durmishi, Rezarta Lalo, Fatjona Kamberi, Shkelqim Hidri, Mitilda Gugu

**Affiliations:** 1Department of Health Care, Faculty of Health, University of Vlora “Ismail Qemali”, 9401 Vlora, Albania; rezarta.petani@univlora.edu.al; 2Scientific Research Centre for Public Health, University of Vlora “Ismail Qemali”, 9401 Vlora, Albania; fatjona.kamberi@univlora.edu.al; 3Faculty of Medical Technical Sciences, “Aleksander Xhuvani” University, 3000 Elbasan, Albania; shkelqim.hidri@uniel.edu.al (S.H.); mitilda.gugu@uniel.edu.al (M.G.)

**Keywords:** maternal lifestyle behaviors, pregnant women, dietary diversity, physical activity, perceived stress, social support, antenatal care, Vlora city

## Abstract

**Background:** Maternal health behaviors during pregnancy are crucial for maternal and fetal outcomes. While global research has explored that demographic, clinical, and psychosocial determinants significantly influence these behaviors, evidence from low- and middle-income countries (LMICs), including Albania, remains limited. This study aims to evaluate psychosocial perceptions and health behaviors related to lifestyle among pregnant women in a local Albanian community in order to identify which are higher risk subgroups that need targeted and tailored antenatal care interventions. **Methods:** This multicenter cross-sectional study included 200 pregnant women attending antenatal clinics from May to August 2024 in Vlora city, Albania. Participants were selected using consecutive sampling based on inclusion criteria. Data were collected through a validated questionnaire composed of five sections: demographic/obstetric data; maternal health behaviors; dietary diversity; physical activity, perceived stress; and social support. Clinical and anthropometric measurements were assessed by trained health professionals during antenatal visits. SPSS version 23.0 and binary logistic regression with *p*-value ≤ 0.05 statistically significant were used for data analysis. **Results:** Mean age was 28.3 ± 6.4 years, 71% employed and 83.5% urban residents. Key unhealthy behaviors included tobacco use (25.5%), alcohol consumption (10.5%), exposure to toxins (15%), and low dietary diversity (32%). We found significant correlations between low dietary diversity and rural residence (Adj OR = 2.48), hypertension (Adj OR = 6.88), and overweight/obesity (Adj OR = 2.33). Tobacco use was associated with unemployment and alcohol use with unemployment and hypertension variables. Low/moderate social support and high perceived stress were significantly related with multiple unhealthy behaviors, such as low dietary diversity, inadequate physical activity and antenatal care. **Conclusions:** Unhealthy nutritional behaviors, tobacco and alcohol use and low physical activity are more prevalent risk factors among pregnant women in Vlora city. Priority should be given to vulnerable groups, including rural residents, pregnant women with low social support, high perceived stress and those with hypertension and obesity. Interventions that integrate psychosocial support and health education into antenatal care services are urgently needed to enhance pregnancy outcomes in Albanian communities.

## 1. Introduction

Unhealthy behaviors and lifestyles during pregnancy remain a major issue of public health due to a significant pregnancy and infant adverse health outcomes [1,2,3]. These behaviors are also contributing to high maternal and infant mortality rates globally [4]. Although the global initiative aims to reduce the maternal mortality ratio (MMR) to less than 70 per 100,000 live births by 2030 [5], Albania, a middle-income Southeast European country, already reporting a relatively low MMR of about 8 per 100,000 live births, continues to face challenges related to infant health. According to the Albanian Institute of Statistics (INSTAT), the infant mortality rate in 2023 was 6.1 deaths per 1000 live births, more than twice the European average of 3 deaths per 1000 [6]. This disparity underscores that achieving low maternal mortality does not necessarily reflect optimal maternal and newborn health, as lifestyle-related risk factors such as poor nutrition, smoking, alcohol consumption, sedentary behavior, unsystematic and insufficient prenatal care continue to threaten both maternal and infant well-being [7]. Identifying modifiable lifestyle factors and vulnerable groups of women who engage in unhealthy habits is essential for designing targeted interventions to improve pregnancy outcomes and reduce preventable infant deaths in Albania. In this context, nutritional behaviors during pregnancy play a crucial role, as poor maternal nutrition has been linked to intrauterine growth restriction, low birth weight, premature delivery, and increased maternal morbidity and mortality [4]. A widespread phenomenon is also the failure to meet the recommended levels of micronutrients among pregnant women, especially in low- and middle-income countries, including Albania, as a result of unhealthy eating habits and a lack of a diverse diet. In addition, many women fail to meet the nutritional intake recommendations provided by the World Health Organization (WHO), placing both maternal and fetal health at considerable risk [8,9]. Enhancing dietary diversity is recognized as one of the most effective approaches to preventing both macro- and micronutrient deficiencies during pregnancy. Therefore, ensuring adequate intake of essential nutrients, such as vitamins, minerals, protein, energy, and fluids, through consumption of a wide variety of food groups is critical. While, it is crucial for pregnant women to consume a wide range of food groups to ensure adequate intake of vital nutrients such as vitamins, minerals, proteins, energy, and fluids [4,8]. Furthermore, a combination of lack of physical activity and eating unhealthy foods during pregnancy is positively related to gestational weight gain and obesity, as has been reported by research. According to that, it has been observed that a good portion of pregnant women are affected by overweight and obesity; in the USA, a prevalence of 42% is reported, in various European countries, 30%, and in populations of Asia, 10% [10]. While smoking during pregnancy is related to higher rates of low birth weight and fetal heart rate abnormalities. It has also been reported that the prevalence of pregnant women who smoke has reached 8.1% at the global level, while in Europe it is reported to be 5.9%. Regarding alcohol consumption by pregnant women, the prevalence at the global level is 9.8%; however, these data differ significantly by country [11,12]. Complications that come from a poor diet and a very passive lifestyle include hypertension and diabetes, which normally place these mothers’ pregnancies in the category of high-risk. Women who are considered to have such a pregnancy experience psychological stress and do not find it easy to adapt to the healthy behaviors that they should apply [13].

High levels of stress during pregnancy have also been shown to significantly impact pregnancy outcomes [14]. In contrast to stress, there is social support that reduces stress and increases adaptation [15]. Perceiving social support during stressful situations may contribute to better health outcomes by shaping the pregnant woman’s perception of threats, thereby reducing anxiety levels and strengthening their coping abilities [16]. A significant relation has been reported between the perceived stress of pregnant women and the presence of social support [13]. In addition, evidence suggests that professionals who work in prenatal care clinics are among the most important in alleviating the psychological and social stress that these women experience, while also promoting the importance of having and maintaining a healthy lifestyle. However, how effective this intervention will be depend also on the social and emotional environment in which these pregnant women are living [17]. Despite the global emphasis on maternal health, there is limited evidence from Albania concerning the lifestyle-related health behaviors of pregnant women and how these relate to perceived psychosocial factors such as stress and social support. From what we know so far, in Albania there are no studies that have analyzed and examined these relationships. Therefore, the present study aims to fill this gap in the literature by evaluating the association between psychosocial perceptions and health behaviors related to lifestyle during pregnancy in a local Albanian community in order to identify more vulnerable subgroups that need tailored antenatal care interventions.

## 2. Materials and Methods

### 2.1. Study Area, Study Design and Study Period

A cross-sectional multicenter study was conducted during the period May–August 2024, including pregnant women attending antenatal clinics in primary healthcare settings in Vlora, a major urban center located in southwestern Albania. Vlora serves as a referral hub for both urban and rural populations from the surrounding region and is characterized by diverse sociodemographic conditions, with women from various economic and educational backgrounds. Due to limited infrastructure, insufficient prenatal care services, and a shortage of trained staff in rural facilities, many pregnant women from neighboring rural areas seek antenatal services in Vlora. Although the majority of participants were urban residents, the study also included women from surrounding rural areas, which should be considered when interpreting the generalizability of the findings. The city has four main urban health centers (HCs No. 1–4) that provide prenatal care for most women in the city and nearby communities. Participants were recruited from three centers (HCs No. 2–4) selected for their high patient volume, availability of specialized maternal health professionals, and representativeness of both urban and peri-urban populations. Health Center No. 1 was excluded because of low patient volume and insufficient related professional capacity. The inclusion of these centers ensured representation of diverse sociodemographic conditions while maintaining standardized antenatal care procedures and consistent data quality, minimizing potential bias from service discrepancies in rural facilities.

### 2.2. Study Population, Sampling Procedure and Sample Size Calculation

Eligible participants were selected using the “consecutive sampling” technique. It was invited to participate any pregnant woman who attended the clinic for prenatal care during the study period if met the inclusion criteria. Each pregnant woman was invited only once, during a scheduled antenatal visit, to avoid any possibility of duplication. This approach is consistent with WHO recommendations for facility-based cross-sectional surveys, ensuring representativeness within the study timeframe [18]. The inclusion criteria were (a) fetal gestational age ≥ 20 weeks, since within this period the mother-fetus bond has already been formed and the foundations of health behaviors have been fully laid; (b) singleton pregnancy, because the extra risk of twin and/or multiple pregnancies affects the results of the study; (c) mothers attending a routine antenatal check-up at the time of enrollment, ensuring accessibility and feasibility of participation; and (d) no experience of stressors in the last 6 months (such as death of some family members because it may have other psychological effects), so that no confounding factors would affect psychological assessment outcomes. The exclusion criteria were (a) incomplete responses or refusal to engage with the questionnaire were not considered in the final analysis; (b) pregnant women who are diagnosed with pathologies that impair active participation in the interview. During the study period, a total of 355 pregnant women were receiving antenatal care at the selected clinics. The Yamane formula n = N/1 + N.e^2^ was used to determine the sample size, n = 355/1 + 355. (0.05)^2^ where n = required sample size; N = population size (N = 197 + 92 + 66 = 355, respectively, for antenatal clinics No. 4, No. 2, and No. 3 in the city of Vlora); e = margin of error (expressed as a decimal). Although the Yamane formula was used to estimate the minimum sample size for this cross-sectional survey, it is acknowledged that this approach is primarily suited for descriptive research. Given the finite source population and the absence of reliable local prevalence data on psychosocial and lifestyle-related factors among pregnant women in Albania, this method was considered appropriate for the exploratory objectives of the study. The calculation was based on a 95% confidence level and a 5% margin of error, resulting in a minimum required sample of 188 participants. Of the 355 women assessed, 200 met all inclusion criteria and were therefore included in the study. To strengthen the methodological rigor beyond descriptive estimation, sample adequacy was further evaluated in relation to the inferential analyses performed. The final sample size (n = 200) was sufficient for multivariable ordinal logistic regression (PLUM), meeting commonly recommended criteria for model stability, including adequate observations per predictor (≥10 events per variable) and full representation of all ordinal outcome categories. Model fit indices and the test of parallel lines further supported the validity and stability of the fitted models.

### 2.3. Data Collection

In this study, a measuring instrument consisting of five sections was used and developed through a structured multi-step process. To enhance the validity of the tool and the reliability of the statements, Sections I to IV are based on various publicly accessible survey instruments, including the Food Frequency Questionnaire [19], the Health Promotion Lifestyle Profile, from which we selected one of its six dimensions to assess the physical activity behaviors of pregnant women [20], the Perceived Stress Scale [21], and the Multidimensional Scale of Perceived Social Support [22]. The items related to maternal health behaviors were adapted from previous research [23,24] and developed specifically for use in Albania. To ensure linguistic and contextual accuracy, the questionnaire was translated into Albanian through a forward and backward translation process by three independent translators. Following the translation, three healthcare professionals with expertise in antenatal care reviewed the instrument and suggested necessary revisions based on their professional judgment. A pilot study with ten pregnant women was conducted to evaluate the questionnaire’s clarity and applicability. Based on the feedback gathered, minor adjustments were made to finalize the questionnaire for use in the present study.

The validation of the questionnaire was conducted through an assessment of internal consistency, evaluated using Cronbach’s alpha. All scales demonstrated alpha coefficients >0.85, indicating strong internal reliability. An Exploratory Factor Analysis (EFA) was performed for the dietary food groups, the physical activity construct, the perceived stress construct, and the social support scale, using Principal Axis Factoring with Varimax rotation. The Kaiser-Meyer-Olkin (KMO) measure ranged from 0.64 to 0.83, and Bartlett’s Test of Sphericity was statistically significant (*p* < 0.001), confirming the adequacy of the data for factor analysis. Factor loadings were ≥0.40 for the majority of items, supporting the construct validity of the measures.

The final instrument contains:Demographic and obstetric data self-reported by study participants.Health behaviors of the pregnant women that also have an impact on fetal well-being and development, such as exposure to unhealthy substances (alcohol, tobacco, toxins, and radiation), unhealthy eating behaviors (folic acid consumption, dietary diversity, pre-pregnancy Body Mass Index (BMI), and weight gain during pregnancy), and inadequate antenatal care (time of the first prenatal visit and total number of prenatal visits compared to the number recommended for the gestational age of the fetus). Radiation exposure was defined as self-reported contact with medical imaging (X-rays, CT scans) and urban electromagnetic sources such as mobile towers and Wi-Fi. This operational definition was considered appropriate for the exploratory aims of the study, and results should be interpreted in this context. Each participant was asked how many weeks pregnant she was when she made her first visit to the doctor at the health center and how many visit she had made in total up to the time she was being interviewed. The categorization of antenatal care contacts into inadequate (<8), adequate (=8), and more than recommended (>8) was based on WHO guidelines for a positive pregnancy experience. The timing of the first antenatal care (ANC) visit was classified into three categories, based also on WHO recommendations: early (<6 weeks), on time (6–12 weeks), and late (≥13 weeks) gestation [25].To measure the dietary diversity of women of reproductive age, an adapted Food Frequency Questionnaire based on Food and Agriculture Organization of the United Nations (FAO) standards was utilized. We used Women’s Dietary Diversity Score (WDDS) which serves to estimate the likelihood of adequate micronutrient consumption through the diversity of food groups consumed. Each participant will be given a list of nine types of food and asked to indicate which of the food items they have consumed in the past 24 h. Dietary diversity scores were determined by adding the number of different food groups consumed by each participant. Based on FAO guidelines, women with dietary diversity scores below the mean score of the sample are considered to have low dietary diversity [19].Physical activity (PA) is internationally recognized as an important factor for protecting and improving health in pregnant women. The physical activity subscale of the Health-Promoting Lifestyle Profile II (HLPL-II), which is a widely used tool in clinical and epidemiological studies including studies among pregnant populations [13], translated and validated in many languages of the world, will be used to assess behaviors related to physical activity. Based on the HPLP II scores, each response ranged from 4 to 32 points for the physical activity subscale. The result will be calculated from an average of the individual’s responses to the 7 items for the physical activity subscale. Permission to use this questionnaire was obtained from the Albanian authors who translated, adapted, and validated it in Albanian [26].Two standardized questionnaires were used to assess psychosocial factors: the Perceived Stress Scale (PSS) [27], based on a Likert scale, and the Multidimensional Scale of Perceived Social Support (MSPSS) [28], which includes 12 items scored on a Likert scale, with a self-report method. Each individual score on the PSS can be a number between 0 and 40, with higher scores symbolizing higher perceived stress levels. Each subscale of the MSPSS has 4 items, and the final score can be a minimum of 12, up to a maximum of 84, with higher scores representing greater social support.Anthropometric measurements of height and weight to determine pre-pregnancy BMI and weight gain during pregnancy was used. Enumerators recorded each respondent’s self-reported pre-pregnancy height and weight. Women were asked to attend fasting (≥8 h) to standardize glucose and blood pressure measurements. Trained midwives performed all anthropometric and clinical assessments for pregnant women using calibrated equipment in selected antenatal centers. The same procedures were used across centers to ensure comparability. We followed WHO guidelines to calculate each participant’s BMI before pregnancy [29]. To calculate the weight gain of the pregnant woman, the weight she had before pregnancy was subtracted from the weight she had at the prenatal visit. According to WHO guidelines [30,31], women were then classified into three categories: below, within, or above the standard range of weight gain. Gestational weight gain was evaluated using the Institute of Medicine (IOM) guidelines, which specify recommended weight-gain ranges for each pre-pregnancy BMI category. Based on these standards, weight gain was classified as below, within, or above the recommended range for each participant. Clinical measurements include measurement of systolic and diastolic blood pressure (BP) using a digital sphygmomanometer in mm Hg, as well as fasting glucose (FG) using a glucometer.

The questionnaire was administered face-to-face within the health centers where these pregnant women participating in the study had routine visits, by initially obtaining verbal informed consent. It was chosen for practical and ethical reasons: (1) the study collected anonymous questionnaire data and non-invasive clinical measurements routinely performed during antenatal visits (blood pressure and fasting glucose), which do not pose significant risk; (2) participant literacy levels varied in this setting, and some women were uncomfortable signing formal documents; thus, verbal consent increased inclusivity and reduced selection bias. The environment where the interview was conducted was private, to allow the woman to feel as comfortable as possible and with the assurance that everything would be confidential. Data collection was performed by three trained antenatal care professionals (two midwives and one nurse) who were not directly responsible for the participants’ subsequent clinical follow-up, in order to reduce social desirability bias. Interviewers received training from the lead researcher not only on how to administer the questionnaire but also on the most ethical and emotionally compassionate way of communicating with the pregnant woman. Each interview lasted approximately 20–25 min.

### 2.4. Statistical Analysis

Data were analyzed using IBM SPSS Statistics for Windows, Version 23.0 (Armonk, NY, USA: IBM Corp.). To present descriptive statistics, cross-tabulations were used. Also, relative and absolute frequencies were reported for categorical variables. While for the continuous variables, mean values and standard deviations were calculated. The Shapiro–Wilk test was used to assess the normality of continuous variables, and all variables showed a non-normal distribution (*p* < 0.05). To control for the risk of Type I errors arising from multiple testing, *p*-values were adjusted using the Bonferroni correction method. Given the presence of several independent statistical tests, *p*-values were adjusted using the formula (*p*/n). Only the *p*-values that remained below the predetermined threshold for statistical significance after adjustment were considered statistically significant.

Variable coding was performed as follows:Dietary Diversity: <6 food groups = Low dietary diversity; ≥6 food groups = High dietary diversity.Physical Activity mean score: ≥2.3 = Adequate; <2.3 = Inadequate.Stress Level mean score: Low (≤2.0); Moderate (2.1–2.8); High (≥2.8).Perceived social support mean score: ≥4.5 indicates high support; <4.5 indicates low support.

Variable selection for multivariate regression: Variables included in the multivariate regression models were selected using a two-step approach. First, bivariate analyses identified factors significantly associated with the outcomes of interest (*p* < 0.05). Second, variables with established theoretical and epidemiological relevance to maternal health behaviors and psychosocial outcomes were retained, even if their bivariate associations were borderline, to ensure adequate control for potential confounding. Multicollinearity among independent variables was assessed using the Variance Inflation Factor (VIF), with all values remaining below the commonly accepted threshold of 5, indicating no significant multicollinearity. To examine associations between binary dependent variables and predictors, univariate and multivariate logistic regression analyses were conducted. Crude odds ratios (ORs) were estimated using Forward selection, whereas adjusted odds ratios.

(AdjORs) were obtained using the Enter method, with all selected covariates entered simultaneously. For ordinal outcomes, ordinal logistic regression (PLUM) was conducted as a sensitivity analysis, and the parallel lines assumption as well as Nagelkerke R^2^ were checked to ensure model adequacy. Effects were expressed as Odds Ratios (OR) and Adjusted Odds Ratios (AdjOR), with 95% confidence intervals (CI) and *p*-values reported to assess precision and statistical significance. For regression purposes, all variables with more than two categories were dichotomized as follows:Educational level: ≤High school (elementary/secondary/high school) vs. >High school (Bachelor’s/Master’s degree)Economic status rate: Low, High, or ModerateParity: Multigravida (second or subsequent pregnancies) versus PrimigravidaSchedule of the first prenatal visit: early (before 13 weeks) versus late (after 13 weeks)BMI: Normal or underweight versus overweight or obese.

## 3. Results

Referring to Table 1, the average age of the participants was 28.3 ± 6.43 years, 71% were employed, and 83.5% resided in urban areas. Approximately 40% of participants had a secondary or lower level of education, and 72.5% reported a moderate socioeconomic status. About one-third of pregnant women (39%) were at their first pregnancy, and 11% had fewer than the recommended number of antenatal visits. Only 19% of pregnant women had their first antenatal visit later than recommended (after 13 weeks). Regarding anthropometric and clinical measurements, the majority of pregnant women (73%) had a normal body weight before pregnancy, 17.5% were overweight, and only 5.5% were obese. About 10.5% had BP over 140/90, and 9.5% of respondents had FG > 140 mg/dL.

Maternal Health Behaviors are presented in Table 2. We observed a significant rate of cannabis and other drug use (5.5%), alcohol use (10.5%), exposure to toxins (15%), and exposure to radiation (17.5%) during pregnancy. Additionally, the results showed higher rates of tobacco exposure, with 25.5% of women using tobacco during pregnancy and 62.5% of them being exposed to tobacco. Regarding supplement intake during pregnancy, the results showed that approximately 72.5% of women reported taking folic acid, while 69.5% had taken a combination of folic acid, vitamin D, and iron supplements. About 35% of women gained weight during pregnancy above the standard weight, while 25% weighed less than the standard. About 32% of women had low dietary diversity, and 69% did not participate in regular physical activity.

Table 3a,b shows the link between unhealthy maternal behaviors and demographic, obstetric, and clinical characteristics. We found that residence, BP over 140/90, and BMI are significantly associated with low dietary diversity. Pregnant women residing in rural areas (Adj OR = 2.48; 95% CI: 1.26–5.80, *p* = 0.001) were more likely to have low dietary diversity, with an effect approximately 2.5 times higher compared to those living in urban areas. Additionally, pregnant women with blood pressure over 140/90 mmHg (Adj OR = 6.88; 95% CI: 2.34–20.69, *p* = 0.001), and overweight/obesity (Adj OR = 2.03; 95% CI: 1.03–3.23, *p* = 0.002) were more likely to have low dietary diversity.

Regarding the unhealthy maternal behavior “weight gain out of standard range,” the findings showed that pregnant women with lower levels of formal education (Adj OR = 1.83; 95% CI: 1.09–3.37; *p* = 0.004) and first-time pregnant women (Adj OR = 3.12; 95% Cl: 1.08–6.02; *p* = 0.001) had increased odds of experiencing weight gain above the standard range. In contrast, age above the mean was significantly associated with lower odds of experiencing weight gain above the standard range (Adj OR = 0.39, 95% CI: 0.21–0.73, *p* = 0.003).

Women’s age and employment status were significantly associated with tobacco exposure. Older women (Adj OR = 1.97; 95% CI: 1.18─3.76, *p* = 0.002) were more likely to be exposed to tobacco, including secondhand smoke, with an effect 2 times higher than among younger women. In contrast, unemployed women was significantly associated with lower odds of tobacco exposure (Adj OR = 0.38, 95% CI: 0.16–0.91, *p* = 0.003).

The findings also showed a significant association between alcohol use, employment status, residence, and BP readings over 140/90. Unemployed pregnant women (Adj OR = 2.59; 95% CI: 1.99–6.75, *p* = 0.002), women from rural areas (Adj OR = 2.21; 95% CI: 1.08–6.33, *p* = 0.003), and those with blood pressure over 140/90 (Adj OR = 4.31; 95% CI: 1.74–15.45, *p* = 0.005) had higher odds of consuming alcohol.

Regarding exposure to toxins like pesticides, our results showed a statistically significant link with variables such as age, BP and number of pregnancy. Older women (Adj OR = 3.05; 95% CI: 1.28–7.29, *p* = 0.006) and blood pressure over 140/90 (Adj OR = 5.55; 95% CI: 2.59–24.20; *p* < 0.0001), had higher odds of exposure to toxins like pesticides. In contrast, women with first pregnancy (Adj OR = 0.25; 95% CI: 0.17─0.94; *p* = 0.002), are lower odds of exposure to toxins like pesticides.

The findings also indicated that pregnant women from rural areas (Adj OR = 0.30; 95% CI: 0.15–0.78; *p* = 0.005) was significantly associated with lower odds of radiation with an effect approximately 3 times lower than that observed among women living in urban areas.

No folic acid intake during pregnancy appears to be statistically linked to unemployed women (Adj OR = 3.08; 95% CI: 1.64–6.12; *p* < 0.0001), who are 3 time more likely not to take this supplement. Additionally, not taking a prenatal vitamin containing folic acid, iron, and vitamin D during pregnancy seems to be statistically associated with a lower level of formal education (Adj OR = 1.85; 95% CI: 1.46–5.17; *p* ≤ 0.001), unemployment (Adj OR = 1.67; 95% CI: 1.07─3.51; *p* = 0.002), and residence in rural area (Adj OR = 4.98; 95% CI: 2.15–11.53; *p* < 0.0001).

Regarding physical activity, the findings showed that women experiencing their first pregnancy (Adj OR = 0.39; 95% CI: 0.21–0.74; *p* = 0.004) were more likely to engage in regular physical activity. Finally, we found that age and economic status were significantly associated with inadequate antenatal care. Older pregnant women (Adj OR = 2.02; 95% CI: 1.04–4.35; *p* = 0.006) and those with low economic status (Adj OR = 6.33; 95% CI: 2.27─17.69; *p* < 0.0001) were more likely to have their first antenatal visit later than recommended and to have fewer total antenatal visits than recommended. Additionally, they were more likely to experience these issues (Adj OR = 3.01; 95% CI: 1.14–12.15); *p* = 0.003) and (Adj OR = 4.81; 95% CI: 1.21–19.18; *p*< 0.0001).

The link between unhealthy maternal behaviors and clinic characteristics, along with social determinants of health, is shown in Table 4. We identified that low dietary diversity (Adj OR = 2.29; 95% CI: 1.23–4.25; *p* = 0.004), incorrect physical activity (Adj OR = 2.57; 95% CI: 1.39–4.89; *p* = 0.003), alcohol consumption (Adj OR = 3.51; 95% CI: 1.22–10.07; *p* = 0.002), toxin exposure (Adj OR = 2.65; 95% CI: 1.14–6.16; *p* = 0.003), radiation (Adj OR = 2.57; 95% CI: 1.14–5.83; *p* = 0.003), lack of prenatal vitamins (Adj OR = 2.51; 95% Cl: 1.32–4.79; *p* = 0.004), inadequate antenatal care (first antenatal visit later than recommended), (Adj OR = 3.49; 95% CI: 1.58–7.71; *p* = 0.002) and failure to follow a pregnancy support program (Adj OR = 3.21; 95% CI: 1.91–6.07; *p* < 0.0001) are significantly connected with low or moderate social support. Moreover, low dietary diversity (Adj OR = 1.85; 95% CI: 1.05–3.51; *p* = 0.003) and BP over 140/90 (Adj OR = 3.01; 95% CI: 1.01–10.52); *p* = 0.001) were significantly linked to high perceived stress during pregnancy.

## 4. Discussion

A total of 200 pregnant women from a local Albanian community participated in the present study. The complexity of mothers’ behaviors and lifestyle, as well as how heavily they are influenced by demographic, clinical, and psychosocial factors, was outlined by the results.

Clinical characteristics of pregnant women.

A prevalence of 10.5% and 9.5%, respectively, of high blood pressure (≥140/90 mmHg) and fasting glucose levels > 140 mg/dL was found, (Table 1). This indicates a considerable burden of gestational hypertension and potential gestational diabetes among the study participants. These results are consistent with existing literature, indicating that hypertensive and glycemic disorders during pregnancy affect 10–22% of pregnant women, and this prevalence is expected to increase in the future as a result of advancing maternal age and the growth of contributing factors such as obesity and metabolic syndrome [32,33].

### 4.1. Unhealthy Lifestyle Behaviors and Their Determinants

The high prevalence of unhealthy behaviors during pregnancy, including tobacco exposure, radiation exposure, exposure to toxins, alcohol use, and cannabis/other drug use, is alarming, (Table 2). These results indicate that tobacco use represents the highest and increasing prevalence among unhealthy behaviors during pregnancy, compared to a previous study conducted in Albania, which reported a smoking rate of 16% among pregnant women. In addition, the high values of tobacco exposure found in the current study are consistent with the literature, where 60–70% of pregnant women are exposed to second-hand tobacco, as a study suggests [34]. Also, as indicated by other research conducted, the association between tobacco exposure and pregnant older women or employed women may result in cumulative exposure in the future and increase exposure to second-hand tobacco in work and other social environments [35].

Surprisingly, the levels of exposure to toxins and radiation were also high (Table 2), which was unexpected because in other studies these results have been much lower [23]. Research indicates that toxins and radiation are well-known teratogens that increase the risk of spontaneous abortion and fetal abnormalities [36]. According to our research, pregnant women who were older and from lower socioeconomic backgrounds were roughly 3 times more likely to be exposed to dangerous chemicals, whereas first-time pregnant women were roughly three times less likely to be exposed than those who had previously been pregnant, (Table 3). This conclusion is expected given since earlier findings indicates that exposure to health-harming chemicals during pregnancy is correlated with age and socioeconomic level [37]. Additionally, we found that women with high BP (>140/90 mmHg) were around 5.5 times more likely than women with normal BP (≤140/90 mmHg) to be exposed to chemicals or pesticides, (Table 3). This strong correlation is consistent with recent research showing that hazardous chemicals affect gestational hypertension. Although more research is required to validate these findings, the evidence presented above points to a possible link between exposure to household and agricultural pollutants, including herbicides, insecticides, and a higher risk of preeclampsia and gestational hypertension [38]. In terms of radiation exposure, our findings indicated that urban women are more likely than their rural counterparts to be exposed to radiation, (Table 3). Urban locations are more likely to have mobile towers, Wi-Fi, and medical imaging facilities like X-ray clinics, all of which emit electromagnetic radiation, thus this is not surprising [39]. However, it should be noted that actual exposure levels were not directly measured in this study, and these associations should be interpreted with caution as they are based on residence rather than quantitative assessment of radiation sources.

Conversely, we found that pregnant women in rural areas were far more likely to drink alcohol, particularly if they were unemployed or had hypertension, (Table 3). This may be connected once again to the fact that health education is less accessible in rural areas, there are fewer creative and social activities available, and alcohol is more readily available and less expensive [40]. This is supported by other research, which demonstrates how cultural norms and a lack of assistance can encourage vulnerable groups to drink alcohol while pregnant [41]. For instance, a study conducted found that pregnant women who drank alcohol were 4.31 times more likely to suffer hypertensive problems [42]. This outcome highlights the urgent need for better, more comprehensive prevention programs in light of all these dangers, particularly for pregnant women who are already experiencing social or economic difficulties.

### 4.2. Nutritional Behaviors and Antenatal Care

A considerable percentage of individuals did not adhere to the recommended prenatal supplementation (30.5%), even though the majority reported using folic acid, (Table 2). Folic acid has been known to help prevent prenatal neural tube defects [43]. Further research, including large clinical trials, has shown and demonstrated that taking folic acid supplements before getting pregnant considerably lowers the risk, particularly in high-risk pregnancies. Only 42.5% of participants in the study reported taking folic acid both before and during pregnancy, despite current guidelines recommending starting it at least 12 weeks prior to conception [44]. Furthermore, almost 30% of women, particularly those from rural areas, did not begin taking supplements until after they found out they were expecting, (Table 2 and Table 3). These results show that there is a substantial knowledge gap and that better education and public health initiatives are required to raise awareness of the significance of preconception folic acid intake among women of reproductive age. According to our research, women who lived in rural areas, had lower economic status, and were unemployed were less likely to take all the recommended prenatal supplements, (Table 3). These results are similar with another study carried out in pregnant women, which showed a correlation between low consumption of micronutrient supplements and socioeconomic difficulties [45].

Also, a substantial proportion of participants demonstrated a low level of dietary diversity, which was highly correlated with low social support, living in a rural area, being obese, having high blood pressure. This group of risk variables reveals a vicious cycle: women who lack dietary variety are also at risk for metabolic problems, most likely as a result of limited availability, affordability, and nutrition knowledge, (Table 2 and Table 3). These trends are in line with earlier research showing that poor diet during the prenatal and periconceptional stages is linked to a number of maternal health issues, such as gestational diabetes and hypertension [46,47]. Low dietary diversity was substantially linked to aberrant weight growth in our study (Table 3), which is consistent with earlier research that links metabolic disturbances during pregnancy to inadequate or unbalanced nutrient intake [23]. Furthermore, fetal growth may be negatively impacted by dietary abnormalities during pregnancy. Fetal growth restriction, changed birthweight, and an increased risk of chronic diseases later in life have all been linked to maternal malnutrition exposure, whether from inadequate or excessive consumption [48]. Primiparity, low educational attainment, and young mother age were all substantially correlated with weight gain that was outside the normal range. These groups might not be aware of the recommended healthy weight growth, and their lack of engagement with prenatal care providers could make the problem worse. Interestingly, weight gain below acceptable levels was also linked to high blood pressure, which raises questions about undiagnosed pregnant hypertensive diseases or nutritional deficiencies [23,49].

According to the results of our study, only 11% of pregnant women reported having fewer than the necessary number of visits, indicating that most of them used antenatal care services. Our findings show that 19% of pregnant women had their first prenatal visit later than advised (Table 1), despite national efforts in Albania to improve access to prenatal care and promote maternal education through collaborations between governmental entities and organizations like the United Nations Population Fund (UNFPA) and World Vision [50]. Women who were older and from poorer socioeconomic backgrounds showed a stronger trend. These findings are consistent with other studies conducted in low- and middle-income (LMIC) nations, where access to maternal health services is hampered by both structural and financial obstacles. Key obstacles to the use of maternal health services in low-income or middle-income economies were recently highlighted by a scoping assessment. These included a lack of information, cultural norms, financial limitations, and restricted access to medical facilities, especially in rural areas. Negative outcomes for both mothers and newborns are linked to delayed or insufficient prenatal care, which is a result of these structural issues [51]. In order to overcome these obstacles, it is imperative that Albanians increase their level of public knowledge and encourage early use of prenatal care.

### 4.3. Psychosocial Determinants: Stress and Social Support

According to our research, pregnant women who had low or moderate social support were significantly more likely to engage in unhealthy behaviors, such as drinking alcohol, eating a diet with little variety, being exposed to toxins or being exposed to radiation, not taking prenatal vitamins, receiving inadequate prenatal care, failure to follow a pregnancy support program and incorrect physical activity (Table 4 and Table 5). These results are consistent with previous research [13], which highlights the protective function of social support during pregnancy. Women who feel a lot of support are more likely to adopt healthier habits, go to prenatal care on a regular basis, and have lower stress levels. The role of social networks in promoting healthy lifestyle choices during pregnancy was further supported by a cross-sectional study that revealed a significant positive correlation between pregnant women’s adoption of health-promoting behaviors and their perception of social support [14].

Additionally, our study found that stress perception during pregnancy was a significant predictor of negative health outcomes, including high blood pressure and low dietary diversity (Table 4 and Table 5). The hypothalamic–pituitary–adrenal (HPA) axis is known to be dysregulated by chronic stress, which can result in changes in cortisol production, cellular immunity, endothelial dysfunction, and hypertension [13]. Meanwhile, another study showed that psychosocial stress during pregnancy can negatively impact the health of both the mother and the fetus by altering cytokine production and raising inflammatory markers [52]. Additionally, our study found a strong correlation between decreased dietary diversity and reported stress during pregnancy. This result was in line with earlier studies showing that psychological stress has a negative impact on women of reproductive age’s nutritional quality [53]. In support of this, a study carried out revealed that women who were more stressed were more likely to follow a “Western-style” diet, which is defined by a low intake of fruits and vegetables and a high intake of processed foods, fats, and sugars. Pregnancy problems may become more likely as a result of this food pattern [54]. These results highlight how crucial it is to incorporate mental health screening into regular prenatal care, particularly for women with poor obstetric histories or little financial resources. Although conducted in a local Albanian community, these findings provide insights relevant for other low- and middle-income countries with similar sociocultural and healthcare challenges, underlining the importance of addressing psychosocial and lifestyle factors during pregnancy globally [13,14].

### 4.4. Strengths and Limitations

This study has some strengths relevant for the respective literature. One of its key strengths is its methodological rigor. The use of validated questionnaires and standardized clinical measures allows for a detailed analysis of both subjective and objective aspects of maternal health. In addition, the sample size and diversity further strengthen the study. A relatively large and diverse sample, spanning different trimesters and socioeconomic backgrounds, enhances the generalizability of the findings to the local pregnant population. Furthermore, the findings of this study are insightful. It identifies links between demographic, clinical, and psychosocial factors and unhealthy behaviors, supporting the development of targeted health promotion strategies.

However, the study also has some limitations. The cross-sectional design limits the ability to draw causal inferences between psychosocial factors and health behaviors and does not allow determination of temporal direction; therefore, reverse causation cannot be excluded. Additionally, reliance on self-reported data may introduce recall bias and social desirability bias, particularly for sensitive behaviors such as smoking and alcohol use, which may be underreported within the local cultural context.

Self-reported assessment of sensitive behaviors, including smoking, alcohol, and drug use, may have led to under-reporting due to social desirability and cultural norms surrounding pregnancy, potentially resulting in conservative prevalence estimates.

Conducting the study in a single urban region further restricts the geographic scope of the results. Therefore, future research should include multiple regions to improve national representativeness. Moreover, the absence of biochemical or physical activity measurements limits the depth of lifestyle behavior assessment.

Despite these limitations, the study’s main strength lies in its novelty and relevance. It is one of the few community-based studies in Albania exploring psychosocial perceptions and lifestyle behaviors during pregnancy. In addition, the findings of this study could serve as a reference baseline for further research in the field.

## 5. Conclusions

This study conducted in pregnant women in a local Albanian community has revealed important insights into the psychosocial perceptions and health behaviors related to lifestyle during pregnancy. Results indicated that unhealthy behaviors, such as low dietary diversity, physical inactivity, exposure to toxic substances, and inadequate prenatal care, are significantly associated with various demographic, clinical, and psychosocial factors, including older mothers, lower levels of formal education and economic status, women experiencing their second or more pregnancy, higher perceived stress, and lower perceived social support. As a consequence, the need for context-specific and multidisciplinary interventions aimed at promoting healthy lifestyles among pregnant women should be underscored. Psychosocial assessments into routine antenatal care and delivering targeted health education programs that address both behavioral and emotional needs should be an integral part of public health strategies. Such strategies may include brief psychosocial screening during standard antenatal visits, targeted health education sessions delivered by midwives and nurses, improved referral pathways to mental health specialists within Family Health Centers, and community-based groups. Finally, fostering the role of healthcare professionals in pregnant maternal community-based health promotion should be more emphasized and fostered. It could be a key factor in reducing preventable pregnancy-related risks and improving maternal and child outcomes.

## Figures and Tables

**Table 1 healthcare-14-00172-t001:** Demographic, obstetrics and clinical characteristics of participants.

Variables	N (%)	Mean ± SD
Sociodemographic variables		
Age (years)		28.30 ± 6.43
Until 20	24 (12.0%)	17.96 ± 1.12
20–30	86 (43.0%)	25.16 ± 2.97
30–40	87 (43.5%)	33.76 ± 3.25
Over 40	3 (1.5%)	41.50 ± 0.71
Employment status		
Housewife	58 (29.0%)	
Employed	142(71.0%)	
Educational status		
Less than high school (8–9 years)	14 (7.0%)	
High school	66 (33.0%)	
Bachelor	80 (40.0%)	
Master	40 (20.0%)	
Residence		
Rural	33 (16.5%)	
Urban	167(83.5%)	
Economic status		
Low	18 (9.0%)	
Average	145 (72.5%)	
High	37 (18.5%)	
Obstetric variables		
Number of pregnancies		
1st	78 (39.0%)	
2nd	74 (37.0%)	
3rd	43 (21.5%)	
4th or more	5 (2.5%)	
Number of antenatal visits		
Less than recommended number	16 (11.0%)	
Equal to recommended number	58 (49.3%)	
More than recommended number	72 (39.7%)	
Time of first antenatal visit		
<6 weeks	52 (26.0%)	
6–13 weeks	110 (55.0%)	
>13 weeks	38 (19.0%)	
Anthropometric and clinical measurements		
BMI (Body Mass Index) before Pregnancy (kg/m^2^)		22.80 ± 3.50
Normal	146 (73.0%)	
Underweight	8 (4.0%)	
Overweight	35 (17.5%)	
Obesity	11 (5.5%)	
Systolic BP (mm Hg)		118.60 ± 21.22
Diastolic BP (mm Hg)		72.50 ± 13.29
BP over 140/90	21 (10.5%)	
FG (mg/dL)		94.70 ± 23.04
FG > 140 mg/dL	19 (9.5%)	148.40 ± 7.11

**Table 2 healthcare-14-00172-t002:** Maternal Health Behaviors.

Variables	N (%)	Mean ± SD
Have you used tobacco, e-cigarettes, vapour in pregnancy?		
Yes	51 (25.5%)	
No	149 (74.5%)	
Are you exposed to second or third-hand smoke, vapour, or other exhaled products in the house or car?		
Yes	125 (62.5%)	
No	75 (37.5%)	
Have you consumed alcohol during pregnancy?		
Yes	21 (10.5%)	
No	179 (89.5%)	
Have you consumed cannabis/other drugs during pregnancy?		
Yes	11 (5.5%)	
No	189 (94.5%)	
Have you been exposed to pesticides or other toxic chemicals during pregnancy?		
Yes	30 (15.0%)	
No	170 (85.0%)	
Have you been exposed to X-rays or other non-medical radiological substances during pregnancy?		
Yes	35 (17.5%)	
No	165 (82.5%)	
Did you just take a folic acid supplement?		
Yes	145 (72.5%)	
Before and during pregnancy	85 (42.5%)	
During pregnancy	60 (30.0%)	
No	55 (27.5%)	
Are you taking a prenatal vitamin with folic acid, iron, and vitamin D?		
Yes	139 (69.5%)	
No	61 (30.5%)	
Dietary diversity		6.54 ± 1.68
Higher dietary diversity (dietary diversity scores > = 6)	136 (68.0%)	7.40 ± 1.16
Low dietary diversity (dietary diversity scores < 6)	64 (32.0%)	4.69 ± 0.99
Weight gain out of standard range	120 (60.0%)	
Below standard range	50 (25.0%)	
Above standard range	70 (35.0%)	
HPLP II– Physical activity subscale		
Correct	62 (31.0%)	
Incorrect	138 (69.0%)	

**Table 3 healthcare-14-00172-t003:** (**a**) Association between unhealthy maternal behaviors and demographic/obstetric/clinic characteristics. (**b**) Association between unhealthy maternal behaviors and demographic/obstetric/clinic characteristics.

(a)																
Variables	Age > Mean				Completed High School (or Below)				Low Economic Status				Unemployment Status			
	OR	Adj OR	95% Confidence Interval for Exp (B)	*p* Value	OR	Adj OR	95% Confidence Interval for Exp (B)	*p* Value	OR	Adj OR	95% Confidence Interval for Exp (B)	*p* Value	OR	Adj OR	95% Confidence Interval for Exp (B)	*p* Value
Low dietary diversity	1.51	1.49	(0.75–3.04)	0.058	1.36	1.21	(0.74–2.49)	0.051	1.39	1.29	(0.51–3.78)	0.513	1.61	1.59	(0.84–3.06)	0.140
Weight gain above standard range	0.37	0.39	(0.21–0.73)	0.003 *	2.04	1.83	(1.09–3.37)	0.004 *	1.2	1.18	(0.44–3.25)	0.717	1.32	1.3	(0.70–2.49)	0.378
Exposure to tobacco (including secondhand smoke)	2.03	1.97	(1.18–3.76)	0.002 **	0.73	0.69	(0.41–1.30)	0.287	1.12	1.07	(0.41–3.03)	0.814	0.5	0.38	(0.16–0.91)	0.003 *
Alcohol consumption	0.58	0.55	(0.23–1.47)	0.253	1.73	1.69	(0.70–4.30)	0.233	0.47	0.44	(0.06–3.77)	0.483	3.08	2.59	(1.99–6.75)	0.002 *
Exposure to radiation	2.25	2.14	(0.87–5.79)	0.143	1.12	1.09	(0.43–2.91)	0.815	0.5	0.47	(0.12–2.03)	0.334	1.05	1.01	(0.38–2.91)	0.922
No taking a prenatal vitamin with folic acid, iron, and vitamin D	1.13	1.11	(0.52–2.49)	0.768	2.78	1.85	(1.46–5.17)	0.001 *	1.85	1.81	(0.51–6.68)	0.347	3.29	1.67	(1.07–3.51)	0.002 *
No folic acid consumption	1.16	1.09	(0.62–2.16)	0.635	0.66	0.58	(0.35–1.25)	0.210	0.43	0.39	(0.16–1.16)	0.149	3.16	3.08	(1.64–6.12)	≤0.0001 *
Exposure to toxins, such as pesticides	3.24	3.05	(1.28–7.29)	0.006 *	1.6	1.54	(0.73–3.49)	0.238	3.29	2.86	(0.95–8.56)	0.051	1.51	1.48	(0.67–3.42)	0.318
Antenatal visits less than recommended number	3.72	3.01	(1.14–12.15)	0.003 *	1.78	1.66	(0.62–5.09)	0.278	5.59	4.81	(1.21–19.18)	≤0.0001 *	1.17	1.15	(0.37–3.67)	0.776
Antenatal visits > 13 week	2.23	2.02	(1.04–4.35)	0.006 *	1.63	1.55	(0.80–3.33)	0.173	6.68	6.33	(2.27–17.69)	≤0.0001 *	1.16	1.11	(0.54–2.50)	0.697
Physical activity (incorrect)	1.6	1.55	(0.87–2.93)	0.128	1.2	1.18	(0.65–2.24)	0.548	0.88	0.79	(0.31–2.48)	0.823	0.79	0.71	(0.41–1.52)	0.497
(**b**)																
**Variables**	**First Pregnancy**				**BMI (Overweight/Obesity)**				**Residence (Village)**				**BP over 140/90**			
	**OR**	**Adj OR**	**95%** **Confidence** **Interval for Exp (B)**	***p* Value**	**OR**	**Adj OR**	**95%** **Confidence** **Interval for Exp (B)**	***p* Value**	**OR**	**Adj OR**	**95%** **Confidence** **Interval for Exp (B)**	***p* Value**	**OR**	**Adj OR**	**95%** **Confidence** **Interval for Exp (B)**	***p* Value**
Low dietary diversity	1.47	1.41	(0.80–2.69)	0.210	2.33	2.03	(1.03–3.23)	0.002 ***	2.71	2.48	(1.26–5.80)	0.001 ***	7.61	6.88	(2.34–20.69)	0.001 ***
Weight gain above standard range	3.55	3.12	(1.08–6.02)	0.001 ***	1.36	1.38	(0.69–2.66)	0.374	1.07	1.01	(0.49–2.33)	0.857	0.87	0.79	(0.22–3.37)	0.851
Exposure to tobacco (including secondhand smoke)	0.56	0.45	(0.31–1.01)	0.402	2.21	1.55	(0.30–7.96)	0.061	1.29	1.22	(0.59–2.80)	0.513	1.61	1.55	(0.54–4.84)	0.389
Alcohol consumption	1.83	1.77	(0.74–4.56)	0.189	2.78	2.64	(0.26–26.60)	0.410	2.94	2.21	(1.08–6.33)	0.003 ***	4.77	4.31	(1.74–15.45)	0.005 ***
Exposure to radiation	2.49	2.22	(0.95–6.55)	0.063	0.36	0.33	(0.04–2.87)	0.331	0.33	0.30	(0.15–0.78)	0.005 ***	0.63	0.6	(0.13–2.96)	0.554
No taking a prenatal vitamin with folic acid, iron, and vitamin D	1.01	1.00	(0.45–2.27)	0.977	1.15	1.11	(0.21–6.56)	0.878	6.56	4.98	(2.15–11.53)	≤0.0001 ***	0.31	0.3	(0.06–1.54)	0.153
No folic acid consumption	1.17	1.11	(0.62–2.21)	0.615	0.55	0.47	(0.25–1.23)	0.146	1.39	1.34	(0.62–3.11)	0.413	0.35	0.33	(0.07–1.70)	0.178
Exposure to toxins, such as pesticides	0.34	0.25	(0.17–0.94)	0.002 ***	1.79	1.77	(0.78–4.17)	0.172	1.68	1.56	(0.65–4.33)	0.278	6.2	5.55	(2.59–24.20)	≤0.0001 ***
Antenatal visits less than recommended number	0.28	0.25	(0.07–1.06)	0.062	2.45	2.33	(0.81–7.47)	0.115	1.01	0.98	(0.26–3.84)	0.989	0.25	0.25	(0.04–1.50)	0.131
Antenatal visits > 13 week	0.49	0.44	(0.22–1.08)	0.079	0.41	0.40	(0.13–1.24)	0.116	2.15	2.05	(0.92–5.03)	0.054	0.39	0.35	(0.04–3.33)	0.391
Physical activity (incorrect)	0.38	0.39	(0.21–0.74)	0.004 ***	1.91	1.88	(0.88–4.24)	0.103	1.49	1.44	(0.63–3.52)	0.361	0.38	0.036	(0.07–15.38)	0.114

Note: (*) significance <0.05, (**) Bonferroni adjusted significance <0.0063. All multivariable logistic regression models were constructed using the same set of predictors (age, education, economic status, employment, parity, BMI, blood pressure, and residence). Multicollinearity was assessed once for this set of predictors, with all VIF values < 2.5, indicating no problematic multicollinearity. Model fit was evaluated separately for each outcome using the Hosmer–Lemeshow test (*p* > 0.05 for all models), while Nagelkerke R^2^ and the area under the ROC curve (AUC) indicated acceptable explanatory and discriminatory power. (***) Bonferroni adjusted significance <0.0063. All multivariable logistic regression models were constructed using the same set of predictors (age, education, economic status, employment, parity, BMI, blood pressure, and residence). Multicollinearity was assessed once for this set of predictors (VIF < 2.5), indicating no significant multicollinearity. Model fit was evaluated separately for each outcome using the Hosmer–Lemeshow test (*p* > 0.05 for all models), while Nagelkerke R^2^ and the area under the ROC curve (AUC) indicated acceptable explanatory and discriminative power.

**Table 4 healthcare-14-00172-t004:** Association between unhealthy maternal behaviors and clinic characteristics with social determinants of health.

Variables	Level of Social Support (Low)				Level of Perceived Stress (High)			
	OR	Adj OR	95% Confidence Interval for Exp (B)	*p* Value	OR	Adj OR	95% Confidence Interval for Exp (B)	*p* Value
Low dietary diversity	2.33	2.29	(1.23–4.25)	0.004 **	1.92	1.85	(1.05–3.51)	0.003 **
Physical activity (Incorrect)	2.62	2.57	(1.39–4.89)	0.003 **	0.81	0.77	(0.28–0.94)	0.053
Weight gain out of standard range	0.59	0.48	(0.30–1.13)	0.112	0.84	0.8	(0.47–1.54)	0.553
Exposure to tobacco	1.45	1.43	(0.82–2.55)	0.197	0.96	0.91	(0.55–1.68)	0.886
Alcohol consumption	3.61	3.51	(1.22–10.07)	0.002 **	1.38	1.31	(0.55–3.43)	0.490
Exposure to toxins, such as pesticides	2.68	2.65	(1.14–6.16)	0.003 **	1.61	1.55	(0.73–3.53)	0.238
Exposure to radiation	3	2.57	(1.14–5.83)	0.003 **	0.81	0.76	(0.39–1.68)	0.577
No folic acid consumption	1.28	1.18	(0.69–2.39)	0.429	0.71	0.65	(0.38–1.31)	0.269
No prenatal vitamins	2.78	2.51	(1.32–4.79)	0.004 **	0.87	0.78	(0.48–1.59)	0.645
Inadequate antenatal care (>13 weeks)	3.5	3.49	(1.58–7.71)	0.002 **	1.94	1.76	(0.93–4.01)	0.074
No prenatal education/classes	1.02	0.99	(0.29–2.55)	0.558	1.82	1.79	(1.02–3.25)	0.042 *
No pregnancy support program	3.44	3.21	(1.91–6.72)	<0.0001 **	1.72	1.65	(0.76–3.89)	0.191
BP over 140/90	0.81	0.78	(0.24–2.76)	0.743	3.27	3.01	(1.01–10.52)	0.001 **
Level of social support (Low)	–	–	–	–	0.88	048	(0.18–1.26)	0.138

Note: (*) significance <0.05, (**) Bonferroni adjusted significance < 0.0042. All multivariable logistic regression models were constructed using the same set of predictors (diet, physical activity, weight gain, smoking, alcohol consumption, exposure to toxins, exposure to radiation, folic acid intake, vitamin supplementation, timing of prenatal visit, prenatal education courses, support programs, blood pressure, and social support—only for the stress outcome). Multicollinearity was assessed once for this set of predictors (VIF < 2.5), indicating no significant multicollinearity. Model fit was evaluated separately for each outcome using the Hosmer–Lemeshow test (*p* > 0.05 for all models), while Nagelkerke R^2^ and the area under the ROC curve (AUC) indicated acceptable explanatory and discriminative power.

**Table 5 healthcare-14-00172-t005:** Sensitivity Analysis.

Variable	Outcome/Predictor	Adj OR	95% CI	*p* Value	GOF *p*	Nagelkerke R^2^	Parallel Lines *p*
Low dietary diversity	BMI (Overweight/Obese)	1.39	(0.52–3.30)	0.153	0.56	0.138	0.229
	BP > 140/90	1.26	(1.10–1.84)	0.025	0.56	0.138	0.229
	Residence (Rural)	2.73	(1.21–5.54)	0.014	0.56	0.138	0.229
Physical inactivity	Age > mean	1.03	(0.42–1.85)	0.939	0.942	0.084	0.914
High perceived stress	BP > 140/90	0.5	(0.16–1.56)	0.233	0.125	0.119	0.085
	Low dietary diversity	1.35	(0.22–3.88)	0.656	0.125	0.119	0.085
Low social support	Physical inactivity	14.14	(1.83–17.45)	0.067	1	0.456	0.007
	Low dietary diversity	1.68	(0.34–10.12)	0.526	1	0.456	0.007
	Alcohol consumption	0.66	(0.21–1.86)	0.433	1	0.456	0.007
	No prenatal vitamin	1.08	(0.43–2.46)	0.855	1	0.456	0.007
	Exposure to radiation	1.03	(0.38–2.68)	0.953	1	0.456	0.007
	First visit > 13 weeks	0.56	(0.20–1.30)	0.173	1	0.456	0.007
	No prenatal course	1.28	(0.1–10.40)	0.942	1	0.456	0.007
	No pregnancy support program	2.0	(0.82–4.88)	0.126	1	0.456	0.007

Note: *p* < 0.05 significance. GOF *p* = 1 indicates no evidence of lack of fit; values are rounded as reported by the software.

## Data Availability

The data presented in this study are available on request from the corresponding author. The data are not publicly available for privacy reasons.

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
