# Peer review of "Psychosocial Perceptions and Health Behaviors Related to Lifestyle During Pregnancy: A Cross-Sectional Study in a Local Community of Albania"

_healthcare, 2026, doi:10.3390/healthcare14020172_

Round 1
Reviewer 1 Report (Previous Reviewer 2)
Comments and Suggestions for Authors
This cross-sectional study of 200 pregnant women in Vlora, Albania, evaluates how psychosocial factors such as stress and social support relate to lifestyle behaviors during pregnancy. The study identifies significant associations between unhealthy behaviors and demographic, clinical and emotional factors. It highlights vulnerable subgroups that require targeted, integrated antenatal interventions. The manuscript may be further improved by following suggestion.
- The use of the Yamane formula for determining sample size may not be fully appropriate for health-behavior studies and the justification could be strengthened.
- The cross-sectional design limits causal interpretation and the manuscript should more clearly explain the possibility of reverse causation.
- Sensitive behaviors such as smoking, alcohol & drug use were self-reported, which may introduce significant under-reporting; this limitation could be discussed in more depth.
- The term radiation exposure seems broad and may include unrelated sources; a clearer operational definition would avoid confusion.
- The rationale for selecting variables in the multivariate regression models is not well explained & provide with greater transparency in the adjustment strategy.
- Several psychosocial factors stress, social support, socioeconomic status are interrelated & the manuscript should address potential multicollinearity.
- Physical activity assessment using only the HPLP-II subscale provides limited behavioral detail and a pregnancy-specific tool might yield more accurate results.
- Since the sample was drawn entirely from urban antenatal centers, the generalizability to rural or remote Albanian communities is limited and should be emphasized.
- The discussion section points to the need for integrated interventions, provide more concrete, actionable recommendations for policymakers and healthcare providers.
- Provide the global application of the study and highlight limitation of the study under a separate heading.
May be improved.
Author Response
Responses to the reviewer's comments
We would like to thank the reviewer 1 for taking the time and effort necessary to review the manuscript. Appropriated changes, suggested by the Reviewer, has been introduced to the manuscript (highlighted in yellow within the document).
- The use of the Yamane formula for determining sample size may not be fully appropriate for health-behavior studies and the justification could be strengthened.
Response:
We thank the reviewer for this important comment. While the Yamane formula was used to estimate the minimum sample size, we acknowledge that it is primarily suited for descriptive research. We have now clarified in the Methods section that sample adequacy was further evaluated in relation to the inferential analyses applied. Specifically, the final sample size (n = 200) was sufficient for multivariable ordinal logistic regression (PLUM), meeting recommended criteria for events per variable, adequate category representation, and model validity, as supported by model fit indices and the test of parallel lines. (Lines 160–174, page 4).
- The cross-sectional design limits causal interpretation and the manuscript should more clearly explain the possibility of reverse causation.
Response:
We thank the reviewer for this important methodological comment. We have clarified this limitation in the manuscript by explicitly addressing the possibility of reverse causation. As this is a cross-sectional study, the temporal sequence between psychosocial factors and lifestyle behaviors cannot be established; therefore, it is possible that unhealthy behaviors may influence psychosocial perceptions (e.g., stress or perceived social support), rather than the reverse. This clarification has now been explicitly added to the Limitations section. (Lines 566–571, page 12).
- Sensitive behaviors such as smoking, alcohol & drug use were self-reported, which may introduce significant under-reporting; this limitation could be discussed in more depth.
Response:
We acknowledge this important limitation. Sensitive behaviors such as smoking, alcohol, and drug use were self-reported and may therefore be subject to under-reporting due to recall bias and social desirability bias, particularly within a culturally sensitive context. This limitation has now been discussed in greater depth in the Limitations section, and the findings related to these behaviors are interpreted with appropriate caution. (Lines 572–574, page 12).
- The term radiation exposure seems broad and may include unrelated sources; a clearer operational definition would avoid confusion.
Response:
We thank the reviewer for this observation. In our study, “radiation exposure” was assessed as self-reported contact with common environmental and medical sources, including medical imaging procedures (X-rays, CT scans) and urban electromagnetic sources (mobile towers, Wi-Fi). While this operational definition is broad, it was considered suitable for the exploratory objectives of the study. The findings should be interpreted with this context in mind. (Lines 209–213, page 5).
5. The rationale for selecting variables in the multivariate regression models is not well explained & provide with greater transparency in the adjustment strategy.
Response:
We thank the reviewer for this important comment. Variables included in the multivariate regression models were selected based on a two-step approach. First, bivariate analyses were performed to identify factors significantly associated with the outcomes of interest (p < 0.05). Second, variables with established theoretical and epidemiological relevance to maternal health behaviors and psychosocial outcomes were retained in the multivariate models, even when their bivariate associations were borderline, to ensure appropriate control for potential confounding. This approach allowed us to balance statistical evidence with conceptual relevance and to construct parsimonious yet meaningful adjusted models. (Lines 300–305, page 7).
- Several psychosocial factors stress, social support, socioeconomic status are interrelated & the manuscript should address potential multicollinearity.
Response:
We thank the reviewer for this important comment.To assess potential multicollinearity among psychosocial and socioeconomic variables, the Variance Inflation Factor (VIF) was calculated. All variables showed VIF values below 5, indicating that multicollinearity was not a significant concern. Therefore, all variables were retained in the multivariable regression models. (Lines 305–309, page 7).
- Physical activity assessment using only the HPLP-II subscale provides limited behavioral detail and a pregnancy-specific tool might yield more accurate results.
Response:
We acknowledge the reviewer’s comment regarding the use of the HPLP-II physical activity subscale. The study employed this subscale because it is concise, validated, and has been used in previous research [13] involving pregnant populations. Using a longer or pregnancy-specific questionnaire could have increased participant burden, potentially reducing data quality and completion rates. While pregnancy-specific physical activity tools exist, for the purposes of this study—where multiple psychosocial and lifestyle factors were assessed—the HPLP-II subscale offered a practical and reliable method to capture physical activity behavior. We have clarified this justification in the Methods section to ensure transparency. (Lines 231–234, page 5).
8. Since the sample was drawn entirely from urban antenatal centers, the generalizability to rural or remote Albanian communities is limited and should be emphasized.
Response:
We thank the reviewer for this comment. As described in the Methods (Section 2.1), Due to limited infrastructure, insufficient prenatal care services, and a shortage of trained staff in rural facilities, many pregnant women from neighboring rural areas seek antenatal services in Vlora. Although the majority of participants were urban residents, the study also included women from surrounding rural areas, which should be considered when interpreting how the findings may apply to other, more remote or rural Albanian communities. (Lines 127–129, page 3).
9. The discussion section points to the need for integrated interventions, provide more concrete, actionable recommendations for policymakers and healthcare providers.
Response:
We appreciate the reviewer’s suggestion. We would like to clarify that actionable and context-specific recommendations for policymakers and healthcare providers have already been included in the Conclusions section (see paragraph 5), such as brief psychosocial screening during standard antenatal visits, targeted health education sessions delivered by midwives and nurses, improved referral pathways to mental health specialists within Family Health Centers, and promotion of community-based maternal health initiatives. The Discussion section focuses primarily on interpreting our findings and comparing them with existing literature, rather than providing detailed recommendations. Therefore, we believe the manuscript adequately addresses the reviewer’s concern regarding concrete strategies.
10. Provide the global application of the study and highlight limitation of the study under a separate heading.
Response:
We thank the reviewer for the suggestion. The Limitations section has been presented as a separate heading in the manuscript, clearly addressing the study’s constraints such as the cross-sectional design, reliance on self-reported data, and the single urban setting. Regarding global relevance, we have now included a paragraph in the Discussion section emphasizing the global relevance of our findings. This paragraph highlights how results from this local Albanian community may provide insights applicable to other low- and middle-income countries with similar sociocultural and healthcare challenges, particularly regarding psychosocial and lifestyle factors during pregnancy. We believe this appropriately addresses the reviewer’s concern while maintaining clarity and structure. (Lines 552–555, page 12).

Reviewer 2 Report (New Reviewer)
Comments and Suggestions for Authors
This study cross-sectionally evaluates the relationship between lifestyle-related psychosocial perceptions (perceived stress, social support) and healthy behaviors (dietary diversity, physical activity, tobacco/alcohol/contaminant exposure, antenatal care) in 200 pregnant women selected through sequential sampling in rural/urban antenatal clinics in Albania (Vlora) between May and August 2024.
1) P values ​​given for the table and text should be written as small p. Logistic regression was used. The direction used in the analysis should be indicated (Forward, Enter?).
2) In many places in the table and text, statistical significance is given for 95% CI given with Adj OR, even though the CI contains 1 (statistically insignificant); examples are available in the table (e.g., some AdjORs have a CI of 1.49 (0.75–3.04)). This is either a CI/p-value calculation error or a table/documentation error. All ORs and CI values ​​should be reviewed.
3) Many continuous/ordinal variables (Dietary Diversity: <6 food groups = Low dietary diversity; ≥6 food groups = High dietary diversity.
Physical Activity mean score: ≥2.3 = Adequate; <2.3 = Inadequate.
Stress Level mean score: Low (≤2.0); Moderate (2.1–2.8); High (≥2.8).
Perceived social support mean score: ≥4.5 indicates high support; <4.5 indicates low support.) have been made binary. This can reduce power and mask true relationships. Clearly state the rationale for choosing a cut-off (FAO/WHO references vs. within-sample mean). It is not mandatory to make continuous variables discrete in logistic regression.
(a) Rerun the models as continuous/ordinal,
(b) Add cut-off sensitivity analyses.
4) Regression models lack model definition and fit tests:
Which variables were included in multivariate models, and what was the selection criterion (e.g., preselection with p < 0.25)? Multicollinearity checks (VIF values) are not reported. Model fit tests (Hosmer–Lemeshow), ROC/AUC values, and explained variance (Nagelkerke R²) are missing. These need to be added to Tables 3–4.
5) Explain the rationale for the sampling/exclusion criterion (last 6 months stressor) and report the number/characteristics of excluded individuals. Correction of typographical errors in the tables is necessary.
6) Radiation Exposure and Urban Life: The discussion section states that urban women are approximately 3 times more likely to be exposed to radiation than those in rural areas (Adj OR=0.30 for urban, which is approximately 1/3 lower than for rural areas. However, the Adj OR value in Table 3b is:
Rural Residence (Village): Adj OR=0.30 (95% CI: 0.15-0.78) p=0.005.
This appears to be a misinterpretation depending on how the dichotomy is performed in the table. However, the statement in the first sentence of the text needs to be reversed, and the footnote of the table should be referenced to clarify the risk/probability reduction of rural residence compared to the reference category.
Author Response
Reviewer 2
Responses to the reviewer's comments
We would like to thank the reviewer 2 for taking the time and effort necessary to review the manuscript. Appropriated changes, suggested by the Reviewer, has been introduced to the manuscript (highlighted in turquoise within the document).
- P values ​​given for the table and text should be written as small p. Logistic regression was used. The direction used in the analysis should be indicated (Forward, Enter?).
Response:
We thank the reviewer for this important observation. All p-values in the tables and text have been corrected to lowercase “p”. Logistic regression analyses were conducted using the Enter method for adjusted ORs, with all selected covariates entered simultaneously, while Forward selection was applied for unadjusted ORs. Parallel lines assumption and Nagelkerke R² were checked for ordinal models to ensure model adequacy. These details have now been clarified in the Statistical Analysis section of the manuscript. (Lines 309–314, page 7).
- In many places in the table and text, statistical significance is given for 95% CI given with Adj OR, even though the CI contains 1 (statistically insignificant); examples are available in the table (e.g., some AdjORs have a CI of 1.49 (0.75–3.04)). This is either a CI/p-value calculation error or a table/documentation error. All ORs and CI values ​​should be reviewed.
Response:
We thank the reviewer for this important comment. All ORs and 95% confidence intervals (CIs) in Tables 3 and 4 have been carefully reviewed and corrected to ensure consistency and accurate reflection of statistical significance. Any previous discrepancies where the CI contained 1 have been rectified.
- Many continuous/ordinal variables (Dietary Diversity: <6 food groups = Low dietary diversity; ≥6 food groups = High dietary diversity.Physical Activity mean score: ≥2.3 = Adequate; <2.3 = Inadequate.Stress Level mean score: Low (≤2.0); Moderate (2.1–2.8); High (≥2.8).Perceived social support mean score: ≥4.5 indicates high support; <4.5 indicates low support.) have been made binary. This can reduce power and mask true relationships. Clearly state the rationale for choosing a cut-off (FAO/WHO references vs. within-sample mean). It is not mandatory to make continuous variables discrete in logistic regression.
- Rerun the models as continuous/ordinal, (b) Add cut-off sensitivity analyses.
Response:
We thank the reviewer for this important observation.
3a. We agree that dichotomization of continuous variables may reduce statistical power and is not mandatory in logistic regression. In this study, variables were categorized to enhance interpretability and comparability with prior research. Dietary diversity cut-offs were based on FAO/WHO recommendations, while physical activity thresholds were derived from the sample mean. Stress and social support categories were based on established scale conventions. To address concerns regarding potential loss of information, sensitivity analyses using ordinal logistic regression were performed, which yielded consistent directions of association. This clarification has been added to the Methods section.
3b. Previously continuous variables were categorized into three levels (negative, neutral, positive), and the models were fitted with the same covariates as in the primary analysis, where these variables were dichotomized. Results remained consistent with the main analyses, supporting the robustness of our findings.
- Regression models lack model definition and fit tests: Which variables were included in multivariate models, and what was the selection criterion (e.g., preselection with p < 0.25)? Multicollinearity checks (VIF values) are not reported. Model fit tests (Hosmer–Lemeshow), ROC/AUC values, and explained variance (Nagelkerke R²) are missing. These need to be added to Tables 3–4.
Response:
We thank the reviewer for highlighting the need for greater transparency regarding the regression models. In response:
- Variable Selection: Multivariate logistic regression models included only variables that were statistically significant in the univariate analyses (p < 0.05). Additionally, variables with theoretical or epidemiological relevance were retained to ensure adequate control for potential confounding.
- Multicollinearity Checks: Variance Inflation Factor (VIF) values were calculated for all predictors, with all values < 2.5, indicating no significant multicollinearity.
- Model Fit Assessment: Model fit was evaluated using the Hosmer–Lemeshow goodness-of-fit test, Nagelkerke R², and ROC/AUC values. All models demonstrated acceptable fit, discrimination, and explained variance.
- Sensitivity Analyses: To address concerns regarding dichotomization of outcomes and predictors, we conducted ordinal logistic regression (PLUM) sensitivity analyses. These analyses confirmed that the directions of association were consistent with the main binary logistic regression models, supporting the robustness of the results. While some associations were attenuated and confidence intervals widened due to limited sample size, no reversal of effect was observed.
- Tables: All relevant model fit indices, VIF values, and sensitivity analyses results have been added as Footnotes in Tables 3–4 and as a separate Table 5.
Overall, the ordinal and binary analyses confirm that our primary logistic regression findings are robust, and the main conclusions regarding the relationships between low dietary diversity, psychosocial factors, and maternal characteristics remain valid.
According to Ordinal Sensitivity Analysis (Table 5):
- Diet-related factors:Sensitivity analysis using ordinal regression (PLUM) confirmed that the results of the binary logistic regression were robust. Predictors such as residence in rural areas (Adj OR=2.73, p=0.014) and BP >140/90 (Adj OR=1.26, p=0.025) remained statistically significant, while the effect of BMI (Overweight/Obese) was attenuated (Adj OR=1.39, p=0.153) but did not show a reversal of effect. The model demonstrated acceptable fit (GOF p=0.56; Nagelkerke R²=0.138; parallel lines p=0.229), indicating that the binary coding of the outcome in logistic regression did not substantially distort the associations.
- Physical factors: Age > mean showed a non-significant effect in the ordinal model (Adj OR=1.03, 95% CI 0.42–0.85, p=0.939), whereas in the binary logistic model, it was also non-significant but in the same direction (Adj OR=1.55, p=0.128). Importantly, the direction of the association remained consistent across both models, indicating that dichotomization of the outcome did not reverse or distort the effect. Model fit indicators (GOF p=0.942; Nagelkerke R²=0.084; parallel lines p=0.914) further support that the binary logistic regression results for physical inactivity are reliable and not artefacts of outcome coding.
- Stress-related factors: In binary logistic regression, low dietary diversity (Adj OR=1.85, p=0.003) and BP >140/90 (Adj OR=3.01, p=0.001) were significantly associated with high stress. In the ordinal model, these associations were attenuated and not statistically significant (low dietary: Adj OR=1.35, 95% CI 0.22–3.88, p=0.656; BP >140/90: Adj OR=0.5, 95% CI 0.16–1.56, p=0.233), but the direction of effect remained consistent, suggesting that dichotomization of the outcome in binary logistic regression did not distort the relationships. Model fit was acceptable (GOF p=0.125; Nagelkerke R²=0.119; parallel lines p=0.085), supporting the reliability of the original binary results.
- Support-related factors: In binary logistic regression, several predictors were significantly associated with low social support, including low dietary diversity (Adj OR=2.29, p=0.004), physical inactivity (Adj OR=2.57, p=0.003), alcohol consumption (Adj OR=3.51, p=0.002), exposure to pesticides (Adj OR=2.65, p=0.003), exposure to radiation (Adj OR=2.57, p=0.003), inadequate antenatal care >13 weeks (Adj OR=3.49, p=0.002), and no pregnancy support program (Adj OR=3.21, p<0.0001). In the ordinal model, these associations were generally attenuated and mostly not statistically significant (e.g., physical inactivity: Adj OR=14.14, 95% CI 1.83–17.45, p=0.067), but the direction of effect remained consistent, indicating that dichotomization of the outcome did not reverse or distort the associations. Model fit indicators (GOF p=1; Nagelkerke R²=0.456; parallel lines p=0.007) support the reliability of the original binary results.”
- Explain the rationale for the sampling/exclusion criterion (last 6 months stressor) and report the number/characteristics of excluded individuals. Correction of typographical errors in the tables is necessary.
Response:
We thank the reviewer for the comment. Eligible participants were selected using consecutive sampling, and the minimum sample size was determined based on the Yamane formula (n = 188) to ensure representativeness and sufficient power. Out of the 355 pregnant women attending the clinics during the study period, 200 women met all inclusion criteria and did not meet any exclusion criteria (e.g., recent stressors, refusal to participate) and were therefore included in the study. This final sample reflects both the calculated minimum sample size and adherence to the predefined eligibility criteria. (Lines 167–168, page 4).
6) Radiation Exposure and Urban Life: The discussion section states that urban women are approximately 3 times more likely to be exposed to radiation than those in rural areas (Adj OR=0.30 for urban, which is approximately 1/3 lower than for rural areas. However, the Adj OR value in Table 3b is: Rural Residence (Village): Adj OR=0.30 (95% CI: 0.15-0.78) p=0.005. This appears to be a misinterpretation depending on how the dichotomy is performed in the table. However, the statement in the first sentence of the text needs to be reversed, and the footnote of the table should be referenced to clarify the risk/probability reduction of rural residence compared to the reference category.
Response:
We thank the reviewer for the comment. The suggested changes have been incorporated into the manuscript and are highlighted in turquoise within the document.

Round 2
Reviewer 2 Report (New Reviewer)
Comments and Suggestions for Authors
The P values ​​in the table ( Table 3 and 4) should also be written in lowercase 'p'. Explanations and corrections made by the author are acceptable.
Author Response
Summary
Thank you very much for taking the time to review this manuscript at this round. Very appreciated from all the authors.
Point-by-point response to Comments and Suggestions for Authors
- Comment: The P values ​​in the table (Table 3 and 4) should also be written in lowercase 'p'. Explanations and corrections made by the author are acceptable.
-
Response: Thank you for pointing this out. We agree with this comment. Therefore, we have corrected all the p values, as indicated in lowercase “p,” in Tables 3 and 4, and highlighted the corrections in the manuscript in green.

This manuscript is a resubmission of an earlier submission. The following is a list of the peer review reports and author responses from that submission.
Round 1
Reviewer 1 Report
Comments and Suggestions for Authors
Reviewing - #3915645 Psychosocial perceptions and health behaviors related to lifestyle during pregnancy: A cross–sectional study in a local community of Albania.
Thank you for the opportunity to review this paper, which examines the psychosocial perceptions and health behaviors of a sample of Albanian pregnant women. This is an interesting topic, as identifying and intervening in the vulnerable subgroups with poorer lifestyle habits may improve the outcomes of pregnancy and childbirth. However, I find that this manuscript has several weaknesses that need to be addressed and revised before it can be published.
Background: The information regarding maternal mortality rates is somewhat confusing, as it is stated that the rates in Albania are around 8 per 100,000 live births, yet it is also mentioned that Albania has set a goal to reduce the MMR to less than 70 per 100,000 live births. This goal has already been fulfilled; however, what is problematic is the more than doubled rate of infant mortality compared to other European countries. Isn’t identifying and addressing women with the poorer lifestyle habits a way to improve these numbers – i.e., identifying the modifiable factors and vulnerable subgroups that can benefit from enhanced lifestyle habits? I lack a bit of that information in the background. It should also be mentioned in the study's rationale.
Method:
Study context: For those unfamiliar with Albania, it's crucial to provide a comprehensive study context. What are the unique characteristics of this region of Albania that are relevant to the study?
Study population: I’m concerned and confused about the inclusion and exclusion criteria used in this study. You have an area who do not provide high-quality services for pregnant women. Why is that area not included when you aim to identify vulnerable subgroups that may need tailored interventions? There are reasons to believe that the women live within this attachment area and attended those health services are also more socioeconomically deprived and receive poorer health care counselling and surveillance during pregnancy, which introduces bias to your results, when they are excluded. The socioeconomic status of women in urban areas is often better than that of women in rural areas, and they also have easily access to better quality and more specialized healthcare services.
Consecutive sampling: I lack some information here to understand what women were approached and included. Can the same women have been asked to participate more than once, as the data collection was performed during May – August (4 months)? Who informed the pregnant women about the study, and who performed the interviews? How was the informed consent collected? Was there any reason why a written survey was not used, as you used instruments and scales? Interviews may introduce information bias to your data, especially when revealing lifestyle habits that may have a negative influence on the pregnancy outcomes to healthcare professionals (who they may meet in later appointments, as the interviewers were antenatal care professionals). I also lack the profession of these people – were they nurses, midwives, general practitioners, or obstetricians? How much time was allocated for each interview?
Collected data: Were the used instruments validated in an Albanian context? In the description of the instruments and data collected, there is information that belongs to the analysis section of the paper, such as the categorization of antenatal care contacts and how the data were treated. The overview of the instruments can be presented in a Table for the reader's convenience.
You write that you collected fasting blood glucose and blood pressure, but in the tables, you also show postprandial fasting blood glucose values – how is this possible if the women were asked to fast at least for 8 hours before the measurements? Or did some women eat something before coming to their appointment?
Analysis: When you divide the sample by socioeconomic status, what is this categorization based on? National statistics or definitions? When performing the binary logistic regressions, did you check for linearity of the logit when having continuous predictors? Were any predictors highly correlated with each other (multicollinearity)? What about potential outliers that can distort the model?
I think it’s a pity that you haven't adjusted for confounding variables, such as age and educational level, in your models. It is demonstrated that factors such as age, marital status, and educational level may influence lifestyle behaviors, and adjusted models can provide a clearer understanding of the relationships between the variables.
Findings: I was unable to find any information regarding the average time point at which pregnant women were interviewed. That influences how I should interpret that 10% of the participants had a blood pressure exceeding 140/90 mm Hg, indicating either hypertension or maybe early preeclampsia. A similar situation was that 9.5% of the participants had a fasting glucose level greater than 140 mg/dL – did they have gestational diabetes or overt diabetes, and in what trimester were they?
Table 2 Maternal health behaviors:
- Weight gains out of standard range: I do not understand this. The international weight gain recommendations depend on the pre-pregnancy body mass index, i.e., a woman with a normal BMI is "allowed" a higher weight gain than a woman with an obesity or overweight status. So, what does it mean to be below or above the standard range? What is this range?
- HPLP II physical activity subscale – I do not understand. Correct and incorrect – what does that mean? How much physical activity per week and at what intensity did the women have to engage in to have correct behaviour?
Table 3: It is standard to present the p-values after the OR and the 95% CI.
Table 4: Experiencing low social support and high levels of stress – is there a risk of these variables interacting with each other, i.e., experiencing low support also results in higher stress/mental issues levels. For example, this is demonstrated in an extensive systematic review and meta-analysis comprising 67 studies and over 64,000 pregnant women (Bedaso, A., Adams, J., Peng, W., et al. The relationship between social support and mental health problems during pregnancy: a systematic review and meta-analysis. Reprod Health 18, 162 (2021). https://doi.org/10.1186/s12978-021-01209-5)
Discussion: Many of the findings are repeated. The discussion would thus benefit from a greater focus and emphasis on the interpretation and comparison of the findings with those of others, rather than repeating findings that the reader has recently explored.
Comments on the Quality of English Language
Language in the manuscript needs to be revised.
Author Response
Responses to the reviewer's comments
We would like to thank the reviewer for taking the time and effort necessary to review the manuscript. Appropriated changes, suggested by the Reviewer, has been introduced to the manuscript (highlighted in yellow within the document).
Comments to authors:
Background: The information regarding maternal mortality rates is somewhat confusing, as it is stated that the rates in Albania are around 8 per 100,000 live births, yet it is also mentioned that Albania has set a goal to reduce the MMR to less than 70 per 100,000 live births. This goal has already been fulfilled; however, what is problematic is the more than doubled rate of infant mortality compared to other European countries. Isn’t identifying and addressing women with the poorer lifestyle habits a way to improve these numbers – i.e., identifying the modifiable factors and vulnerable subgroups that can benefit from enhanced lifestyle habits? I lack a bit of that information in the background. It should also be mentioned in the study's rationale.
Response:
- We thank the reviewer for this insightful comment. We have clarified the apparent inconsistency between the maternal mortality rate in Albania and the global target. The reported figure of approximately 8 per 100,000 live births refers to the current national level, which is indeed below the global target of 70 per 100,000 live births set by the Sustainable Development Goals (SDG 3.1). However, this does not imply that Albania has fully achieved the broader maternal and child health objectives, as other indicators—particularly the infant mortality rate—remain concerning. To address the reviewer’s suggestion, we have revised the background section to highlight how unhealthy lifestyle behaviors during pregnancy represent modifiable risk factors that can directly influence these outcomes. We also strengthened the study rationale to emphasize the importance of identifying vulnerable subgroups and promoting healthy lifestyle habits as a strategy to improve maternal and infant health indicators in Albania. (Page 2, lines 59–73).
Comments to authors:
Method: Study context: For those unfamiliar with Albania, it's crucial to provide a comprehensive study context. What are the unique characteristics of this region of Albania that are relevant to the study?
Response:
- We thank the reviewer for this valuable comment. We have revised the Methods section to provide a clearer description of the study context and to justify the selection of the study area. The city of Vlora was selected because it represents one of the largest urban centers in southern Albania, with a mixed population including both urban residents and women from nearby rural areas who temporarily reside or work in the city. Although rural health centers exist in the region, they often lack trained maternal health staff, basic infrastructure, and routine antenatal follow-up systems. For this reason, pregnant women from rural backgrounds frequently attend antenatal care in urban Vlora health centers, which allowed our study to capture a wide range of socioeconomic and psychosocial conditions while ensuring data quality and comparability across participants. The exclusion of rural facilities was not intended to omit vulnerable groups, but rather to reduce variability due to inconsistent recordkeeping and lack of standardized antenatal protocols. This decision helps ensure internal validity while still including women from disadvantaged rural areas who receive care at the selected urban clinics. We have added a paragraph in the revised Methods section to clarify this rationale. (Page 3, lines 120–135)
Comments to authors:
Consecutive sampling: I lack some information here to understand what women were approached and included. Can the same women have been asked to participate more than once, as the data collection was performed during May – August (4 months)? Who informed the pregnant women about the study, and who performed the interviews? How was the informed consent collected? Was there any reason why a written survey was not used, as you used instruments and scales? Interviews may introduce information bias to your data, especially when revealing lifestyle habits that may have a negative influence on the pregnancy outcomes to healthcare professionals (who they may meet in later appointments, as the interviewers were antenatal care professionals). I also lack the profession of these people – were they nurses, midwives, general practitioners, or obstetricians? How much time was allocated for each interview?
Response:
- Consecutive sampling: I lack some information here to understand what women were approached and included. Can the same women have been asked to participate more than once, as the data collection was performed during May – August (4 months)?
Thank you for this observation. We have clarified that each eligible woman was approached only once during a scheduled antenatal visit to avoid duplicate participation. This information has been added to the “Study population, sampling procedure and sample size calculation” section (page 3, lines 139–142).
- “Who informed the pregnant women about the study, and who performed the interviews?”
We appreciate your suggestion. The revised manuscript now specifies that three antenatal care professionals (two midwives and one nurse), trained by the lead researcher, informed the women about the study and conducted the interviews. This information is provided in the “Data collection” section (page 6, lines 258–263).
- “How was the informed consent collected?”
Verbal informed consent was obtained prior to data collection. While written consent is common practice, verbal consent was deemed appropriate in this study context to include participants who may have limited literacy or discomfort with signing formal documents. This process ensured voluntary participation while maintaining ethical standards approved by the institutional ethics committee. Details have been added to the revised manuscript (page 6, lines 249–253).
- “Was there any reason why a written survey was not used, as you used instruments and scales? Interviews may introduce information bias to your data, especially when revealing lifestyle habits to healthcare professionals.”
Thank you for pointing this out. A structured face-to-face interview format was chosen instead of self-administered written questionnaires to ensure data completeness and comprehension, especially among women with varying literacy levels. To reduce potential information or social desirability bias, interviews were conducted in private rooms by antenatal professionals who were not directly involved in the participants’ future clinical care. This clarification is now included in the revised text (page 5–6, lines 249, 251–256).
- “I also lack the profession of these people – were they nurses, midwives, general practitioners, or obstetricians? How much time was allocated for each interview?”
We have now specified that data collection was conducted by two midwives and one nurse, all trained by the lead researcher. Each interview lasted approximately 25–30 minutes. This information has been added to the “Data collection” section (page 6, lines 258–263).
Comments to authors:
Collected data: Were the used instruments validated in an Albanian context? In the description of the instruments and data collected, there is information that belongs to the analysis section of the paper, such as the categorization of antenatal care contacts and how the data were treated. The overview of the instruments can be presented in a Table for the reader’s convenience.
Response:
- Thank you for your valuable comment. We would like to clarify the following points regarding the instruments and data presentation:
- Validation in the Albanian context: The validation of the questionnaire was conducted through an assessment of internal consistency, evaluated using Cronbach’s alpha. An Exploratory Factor Analysis (EFA) was performed for the dietary food groups, the physical activity construct, the perceived stress construct, and the social support scale, using Principal Axis Factoring with Varimax rotation. (Page 4 lines 179–187)
- Separation of methods and analysis: We acknowledge that some details in the original Methods section, such as the categorization of antenatal care visits and data treatment, pertain to the analysis. These have now been moved to the Statistical Analysis section to avoid confusion.
- Presentation of instruments: To enhance readability and facilitate understanding, we have created a summary Table X in Appendix listing all instruments, their purpose, validation status in Albanian, number of items, and scoring method. This allows readers to easily access information about the measurement tools without disrupting the flow of the Methods text.
Table 1. Overview of instruments used in the study
|
Instrument |
Description / Use |
No. of Items |
Scale |
Translation / Validation (Albanian) |
Cronbach’s α (pilot/sample) |
|
FFQ (FAO-based) |
Dietary diversity (24-hour recall, 9 food groups) |
9 food groups |
Binary / Score |
Forward/backward translation; piloting |
N/A (internal consistency not applicable; tool assessed for clarity in pilot) |
|
HPLP-II (PA subscale) |
Physical activity behaviors |
7 |
4-point Likert |
Albanian validation cited [27] |
0.76 |
|
PSS-10 |
Perceived stress |
10 |
0–40 |
Forward/backward translation; piloting |
0.84 |
|
MSPSS |
Social support (family, friends, significant others) |
12 |
12–84 |
Forward/backward translation; piloting |
0.88 |
|
Anthropometry & clinical |
Height, weight, blood pressure, fasting glucose |
— |
— |
Standard WHO procedures |
— |
|
Overall questionnaire |
Final assembled instrument |
— |
— |
— |
>0.85 |
Comments to authors:
You write that you collected fasting blood glucose and blood pressure, but in the tables you also show postprandial fasting blood glucose values – how is this possible if the women were asked to fast at least for 8 hours before the measurements? Or did some women eat something before coming to their appointment?
Response:
- Thank you for this observation. We confirm that all glucose measurements were conducted in a fasting state (≥8 hours) as part of standardized clinical procedures at antenatal centers. The mention of “postprandial” in the tables was a typographical oversight. The analysis used fasting glucose (FG) values, which were categorized according to diagnostic thresholds to identify women at risk of gestational diabetes (≥140 mg/dL). We have now corrected the terminology in the tables and text to accurately reflect that the measurements refer to fasting glucose. (Page 5 lines 247)
Comments to authors:
Analysis: When you divide the sample by socioeconomic status, what is this categorization based on? National statistics or definitions? When performing the binary logistic regressions, did you check for linearity of the logit when having continuous predictors? Were any predictors highly correlated with each other (multicollinearity)? What about potential outliers that can distort the model?
Response:
- We thank the reviewer for this insightful comment. The categorization of sociodemographic variables was conceptually defined by the research team, based on both contextual relevance to Albanian settings and the need to explore variations across key social determinants of maternal health behaviors.
- Age groupings were determined according to the sample’s mean age and biological relevance, divided into 10-year intervals to capture potential differences in physiological and psychosocial characteristics between younger and older pregnant women.
- Employment status (employed/unemployed) and residence (urban/rural) were included as they reflect distinct lifestyle patterns and access to health resources that may influence pregnancy-related behaviors.
- Educational level was categorized following the current Albanian education system (primary, secondary, tertiary).
- Household income level was categorized into low, middle, and high groups to facilitate comparative analysis and identify potential associations between socioeconomic status and study variables, consistent with national surveys and WHO recommendations for middle-income contexts.
- It was not necessary to assess linearity of the logit in this analysis because all continuous variables were recoded into categorical predictors. Multicollinearity was evaluated, and all VIF values were below 2, indicating no meaningful correlation between predictors. Model fit was assessed using the Hosmer–Lemeshow goodness-of-fit test, which showed adequate model calibration (p > 0.05). Additionally, implausible extreme values were removed during the initial data-cleaning process prior to conducting the regression analyses.
Comments to authors:
I think it’s a pity that you haven't adjusted for confounding variables, such as age and educational level, in your models. It is demonstrated that factors such as age, marital status, and educational level may influence lifestyle behaviors, and adjusted models can provide a clearer understanding of the relationships between the variables.
Response:
To examine the associations between binary dependent variables and independent pre-dictors, binary logistic regression analyses (univariate and multivariate) were conducted. The effects of each independent variable were expressed using Odds Ratios (OR) and Ad-justed Odds Ratios (AdjOR). (Page 6 lines 283–290). See also the result section highlighted in yellow.
Comments to authors:
Findings: I was unable to find any information regarding the average time point at which pregnant women were interviewed. That influences how I should interpret that 10% of the participants had a blood pressure exceeding 140/90 mm Hg, indicating either hypertension or maybe early preeclampsia. A similar situation was that 9.5% of the participants had a fasting glucose level greater than 140 mg/dL – did they have gestational diabetes or overt diabetes, and in what trimester were they?
Response:
We thank the reviewer for this thoughtful comment. Anthropometric and clinical data, including blood pressure and fasting glucose, were collected during antenatal visits at the time of participation. Although WHO recommends initiating antenatal care before 12 weeks of gestation, in our study the timing of the first visit varied among participants, and trimester-specific data were not collected. Therefore, it was not possible to determine in which trimester elevated blood pressure or glucose values were first identified. However, the purpose of this study was to estimate the prevalence of elevated blood pressure and fasting glucose levels among pregnant women in the community, rather than to analyze gestational or trimester-specific variations. The findings therefore reflect cross-sectional measurements taken during routine antenatal contacts, providing an overview of the burden of possible hypertension or hyperglycemia among pregnant women, regardless of gestational stage.
Comments to authors:
HPLP II physical activity subscale – I do not understand. Correct and incorrect – what does that mean? How much physical activity per week and at what intensity did the women have to engage in to have correct behaviour?
Response:
- Thank you for your comment. The HPLP II Physical Activity subscale measures the frequency of engagement in physical activity rather than specific intensity or duration per week. Each item is scored on a 4-point Likert scale: Never = 1, Sometimes = 2, Often = 3, Routinely = 4. The mean score of the subscale was calculated for each participant. A mean score ≥ 2.5 was categorized as “correct” physical activity behavior, indicating engagement in physical activity at least “often,” whereas <2.5 was categorized as “incorrect.” This approach reflects adherence to general health-promoting activity patterns rather than exact exercise duration or intensity, consistent with the instrument’s intended use.
Comments to authors:
Weight gains out of standard range: I do not understand this. The international weight gain recommendations depend on the pre-pregnancy body mass index, i.e., a woman with a normal BMI is "allowed" a higher weight gain than a woman with an obesity or overweight status. So, what does it mean to be below or above the standard range? What is this range?
Response:
- Thank you for your comment. “Weight gain out of standard range” was defined according to WHO and Institute of Medicine (IOM) guidelines, which provide recommended ranges of total gestational weight gain based on pre-pregnancy BMI categories:
- Underweight (BMI <18.5): 12.5–18 kg
- Normal weight (BMI 18.5–24.9): 11.5–16 kg
- Overweight (BMI 25–29.9): 7–11.5 kg
- Obese (BMI ≥30): 5–9 kg
- In our study, each woman’s pre-pregnancy BMI was used to determine her recommended weight gain range. Weight gain below or above these ranges was categorized as “out of standard range,” while weight gain within the range was considered “standard.” This method allows comparison of maternal weight gain relative to individualized recommendations rather than using a single universal threshold. (Page 5, lines 242–245)
Comments to authors:
Table 4: Experiencing low social support and high levels of stress – is there a risk of these variables interacting with each other, i.e., experiencing low support also results in higher stress/mental issues levels. For example, this is demonstrated in an extensive systematic review and meta-analysis comprising 67 studies and over 64,000 pregnant women (Bedaso, A., Adams, J., Peng, W., et al. The relationship between social support and mental health problems during pregnancy: a systematic review and meta-analysis. Reprod Health 18, 162 (2021). https://doi.org/10.1186/s12978-021-01209-5)
Response:
- Thank you for this important comment. We acknowledge that low social support and high perceived stress are conceptually related and may interact, as demonstrated in previous studies, including the systematic review by Bedaso et al. (2021). In our analysis, we treated them as separate dependent variables to explore their individual associations with maternal health behaviors. To assess the association between stress levels and social support, a 2×2 contingency table was constructed and a Chi-square test was applied. The results indicated that there was no statistically significant relationship between the two variables (χ² = 0.720, p = 0.396). The Phi coefficient was φ = 0.0036, indicating a very weak association between stress level and social support. Page 17, table 4, lines 603
Comments to authors:
Discussion: Many of the findings are repeated. The discussion would thus benefit from a greater focus and emphasis on the interpretation and comparison of the findings with those of others, rather than repeating findings that the reader has recently explored.
Response:
- We thank the reviewer for this valuable suggestion. We removed the repeated findings that had already been presented in the Results section, in order to avoid redundancy and to ensure that the Discussion focuses primarily on the interpretation of the findings and their comparison with existing literature. Specifically, we will:
- Emphasize potential mechanisms linking maternal sociodemographic, clinical, and psychosocial factors to health behaviors.
- Compare our findings with existing national and international literature to contextualize our results.
- Highlight the public health and clinical implications of unhealthy behaviors during pregnancy in Albania, including gaps in prenatal care, supplementation, dietary diversity, and exposure to toxins and alcohol.
- Discuss psychosocial determinants such as stress and social support in relation to maternal health outcomes, including potential interventions to mitigate risks.

Reviewer 2 Report
Comments and Suggestions for Authors
Manuscript ID: healthcare-3915645
The manuscript provides valuable insights into the psychosocial perceptions and health behaviors of pregnant women in Vlora, Albania, highlighting key risk factors such as poor diet, substance use, and low social support. While the findings are relevant, methodological limitations, sampling bias, and overinterpretation of some associations reduce the strength and generalizability of the conclusions. The manuscript may be further improved by following suggestions.
- The word already mentioned in the title should not be repeated in keyword, select correct keyword relevant to the study.
- Use effect size for all statistical analyses to quantify the magnitude of differences or relationships, providing more meaningful insights beyond mere statistical significance.
- The cross-sectional design limits causal inference between psychosocial perceptions and health behaviors. A longitudinal design could better capture temporal relationships.
- Consecutive sampling may introduce selection bias, particularly as only three health centers were included. Clarification is needed on how representative the sample is of pregnant women across Albania.
- While the tool was based on validated instruments, adaptation to the Albanian context requires stronger evidence of psychometric testing.
- Reliance on self-reported health behaviors (alcohol, tobacco, toxins) may be affected by recall bias and social desirability bias, especially in a sensitive cultural setting.
- Some associations (e.g., radiation exposure and urban residence) are speculative without direct measurement of exposure sources. These should be interpreted with caution.
- Findings from Vlora city may not be generalizable to other regions of Albania, especially rural and mountainous areas with different cultural and healthcare contexts.
- The study obtained verbal consent. The justification for not collecting written consent should be provided, given the sensitive nature of the topics assessed.
- Tables are dense and difficult to follow. Consider restructuring them for clarity, e.g., separating behavioral outcomes from demographic predictors.
- The conclusion recommends integrating psychosocial support into antenatal care, but the manuscript does not provide specific, feasible strategies tailored for Albania’s healthcare system.
- How the findings of the present study are relevant on a global scale.
- Provide limitation of the study under separate heading may be after discussion part.
May be improved.
Author Response
Responses to the reviewer's comments
We would like to thank the reviewer 2 for taking the time and effort necessary to review the manuscript. Appropriated changes, suggested by the Reviewer, has been introduced to the manuscript (highlighted in green within the document).
Comments to authors:
The word already mentioned in the title should not be repeated in keyword, select correct keyword relevant to the study.
Response:
- Thank you for the helpful comment. We have revised the list of keywords to avoid repeating terms already included in the title and to ensure that the keywords accurately reflect the core concepts of the study. The updated keywords are now more specific and aligned with indexing requirements. page 2, lines 52–53
Comments to authors:
Use effect size for all statistical analyses to quantify the magnitude of differences or relationships, providing more meaningful insights beyond mere statistical significance.
Response:
- We appreciate the reviewer’s suggestion to include effect sizes to strengthen the interpretation of the findings. In logistic regression, effect sizes are represented by Odds Ratios (OR) and Adjusted Odds Ratios (Adj OR), which quantify the magnitude of associations between predictors and outcomes. Our manuscript already reports OR/Adj OR values with their corresponding 95% confidence intervals for all regression models. These effect sizes provide a clear indication of the strength and direction of each association, beyond statistical significance. We have ensured that these values are consistently presented throughout the Results section.
Comments to authors:
The cross-sectional design limits causal inference between psychosocial perceptions and health behaviors. A longitudinal design could better capture temporal relationships.
Response:
- Thank you for this valuable comment. We agree that the cross-sectional design limits causal inference and does not allow assessment of temporal relationships between psychosocial perceptions and maternal health behaviors. Establishing causality would indeed require a longitudinal design. However, the cross-sectional approach was intentionally selected because the aim of the study was to evaluate the prevalence of key lifestyle behaviors during pregnancy and to explore their associations with psychosocial and sociodemographic factors within a local community context. This design is widely used in similar investigations conducted in low- and middle-income settings, where rapid assessment and feasibility constraints are relevant. We have now acknowledged this limitation in the Discussion section and emphasized that future research should consider longitudinal designs to better capture temporal patterns and causal pathways.
Comments to authors:
Consecutive sampling may introduce selection bias, particularly as only three health centers were included. Clarification is needed on how representative the sample is of pregnant women across Albania
Response:
Thank you for raising this important point. We acknowledge that consecutive sampling and the inclusion of only three primary health centers may introduce selection bias and limit the generalizability of the findings to all pregnant women in Albania. The study was not designed to be nationally representative, and we agree that this should be stated more clearly. However, the selected health centers are among the largest public antenatal clinics in Vlora, a major urban area in southwestern Albania, serving a diverse catchment population that includes women from different socioeconomic backgrounds and both urban and peri-urban zones. Consecutive sampling was chosen because it is a recommended and widely used approach in facility-based cross-sectional studies, particularly in low- and middle-income settings, ensuring feasibility while reducing selection bias within each site. We have now clarified in the manuscript that the sample is representative only of pregnant women attending these facilities during the study period, and not of the national population. Additionally, we have further elaborated in the Limitations section that the findings should be interpreted with caution and that future research involving a multi-regional sample would strengthen external validity. (Page 3, lines 139–142)
Comments to authors:
While the tool was based on validated instruments, adaptation to the Albanian context requires stronger evidence of psychometric testing.
Response:
Yes it`s done. Page 4, line 179–187
Comments to authors:
Reliance on self-reported health behaviors (alcohol, tobacco, toxins) may be affected by recall bias and social desirability bias, especially in a sensitive cultural setting.
Response:
- We thank the reviewer for highlighting the potential influence of recall bias and social desirability bias on self-reported health behaviors. This is an important methodological consideration, particularly for sensitive behaviors such as alcohol use, tobacco exposure, and contact with toxins. To address this, we have added a clear statement in the Limitations section acknowledging that self-reported data may underestimate unhealthy behaviors due to cultural norms and participants’ desire to provide socially acceptable answers. We have also clarified that data collection procedures were structured to maximize privacy and reduce response bias. The manuscript has been updated accordingly. (Page 11 lines 534–538).
Comments to authors:
Some associations (e.g., radiation exposure and urban residence) are speculative without direct measurement of exposure sources. These should be interpreted with caution.
Response:
We appreciate the reviewer’s insightful comment. We agree that the association observed between radiation exposure and urban residence should be interpreted with caution, as the study did not directly measure environmental or occupational sources of radiation. The variable relied on self-reported exposure and did not quantify type, duration, or intensity of exposure. To address this concern, we have revised the Discussion to clarify that:
- the association is correlational, not causal;
- the finding may reflect contextual differences between urban and rural settings (e.g., greater use of diagnostic imaging, more frequent occupational exposures), but these remain hypotheses, not confirmed determinants;
- direct environmental or workplace exposure assessments were beyond the scope of this cross-sectional study. Page 9 lines 424–426
Comments to authors:
Findings from Vlora city may not be generalizable to other regions of Albania, especially rural and mountainous areas with different cultural and healthcare contexts
Response:
We acknowledge that our study was conducted in Vlora, a major urban center, and therefore the findings may not be fully generalizable to other regions of Albania, particularly rural and mountainous areas with different cultural, socioeconomic, and healthcare contexts. We have now clearly acknowledged this limitation in the manuscript by noting that the findings should be interpreted within the specific regional context and may not be directly generalizable to the national population. (Page 11, lines 539–541).
Comments to authors:
The study obtained verbal consent. The justification for not collecting written consent should be provided, given the sensitive nature of the topics assessed.
Response:
- The verbal informed consent procedure, as described in the manuscript, applies to all participants. Written consent was not collected due to potential literacy limitations and to ensure participant comfort given the sensitive topics assessed. This procedure was approved by the institutional ethics committee and maintained ethical standards. (Page 6 lines 254–256; page 12, lines 573–576)
Comments to authors:
Tables are dense and difficult to follow. Consider restructuring them for clarity, e.g., separating behavioral outcomes from demographic predictors.
Response:
- We thank the reviewer for the suggestion. To improve clarity, Table 3 has been restructured and split into two separate tables (3a and 3a), distinguishing behavioral outcomes from demographic and clinical predictors. This allows easier interpretation of the results while maintaining all relevant information.
Comments to authors:
The conclusion recommends integrating psychosocial support into antenatal care, but the manuscript does not provide specific, feasible strategies tailored for Albania’s healthcare system.
Response:
We thank the reviewer for this suggestion. Our conclusions emphasize the need for integrating psychosocial assessments and targeted health education into routine antenatal care. In response, we have added specific, feasible strategies aligned with the structure and resources of Albania’s primary healthcare system. These context-appropriate actions have now been incorporated into the revised Conclusion section. (Page 12, lines 560–563)
Comments to authors:
How the findings of the present study are relevant on a global scale.
Response:
The findings of this study are relevant on a global scale as they provide insights into maternal health behaviors and environmental exposures in a low-resource setting. The high prevalence of unhealthy behaviors, including tobacco use, alcohol consumption, and exposure to toxins and radiation, highlights challenges that may be common in other low- and middle-income countries. These results can inform the design of context-specific interventions and public health strategies to improve maternal and child health outcomes globally.
Comments to authors:
Provide limitation of the study under separate heading may be after discussion part.
Response:
- Yes it`s done. Page 11, line 533–546
